# Spatiotemporal dynamics of PIEZO1 localization controls keratinocyte migration during wound healing

Jesse R Holt[1,2,3], Wei-Zheng Zeng[4†], Elizabeth L Evans[1,2†], Seung-Hyun Woo[4†], Shang Ma[4], Hamid Abuwarda[1,2], Meaghan Loud[4], Ardem Patapoutian[4*], Medha M Pathak[1,2,3,5*]

[1]Departmentof Physiology & Biophysics, UC Irvine, Irvine, United States; [2]Sue and Bill Gross Stem Cell Research Center, UC Irvine, Irvine, United States; [3]Center for Complex Biological Systems, UC Irvine, Irvine, United States; [4]Howard Hughes Medical Institute, Department of Neuroscience, The Scripps Research Institute, La Jolla, United States; [5]Department of Biomedical Engineering, UC Irvine, Irvine, United States

*For correspondence:
ardem@scripps.edu (AP);
medhap@uci.edu (MMP)

†These authors contributed equally to this work

Competing interest: The authors declare that no competing interests exist.

**Abstract** Keratinocytes, the predominant cell type of the epidermis, migrate to reinstate the epithelial barrier during wound healing. Mechanical cues are known to regulate keratinocyte re-epithelialization and wound healing; however, the underlying molecular transducers and biophysical mechanisms remain elusive. Here, we show through molecular, cellular, and organismal studies that the mechanically activated ion channel PIEZO1 regulates keratinocyte migration and wound healing. Epidermal-specific *Piezo1* knockout mice exhibited faster wound closure while gain-of-function mice displayed slower wound closure compared to littermate controls. By imaging the spatiotemporal localization dynamics of endogenous PIEZO1 channels, we find that channel enrichment at some regions of the wound edge induces a localized cellular retraction that slows keratinocyte collective migration. In migrating single keratinocytes, PIEZO1 is enriched at the rear of the cell, where maximal retraction occurs, and we find that chemical activation of PIEZO1 enhances retraction during single as well as collective migration. Our findings uncover novel molecular mechanisms underlying single and collective keratinocyte migration that may suggest a potential pharmacological target for wound treatment. More broadly, we show that nanoscale spatiotemporal dynamics of Piezo1 channels can control tissue-scale events, a finding with implications beyond wound healing to processes as diverse as development, homeostasis, disease, and repair.

## Introduction

The skin, the largest organ of the body, serves as a barrier against a myriad of external insults while also performing important sensory and homeostatic functions. Cutaneous wounds interfere with all these functions and expose the body to an increased risk of infection, disease, and scar formation (*Evans et al., 2013*). During the repair of wounded skin, the migration of keratinocytes from the wound edge into the wound bed plays an essential step in re-establishing the epithelial barrier and restoring its protective functions (*Kirfel and Herzog, 2004*; *Gantwerker and Hom, 2011*). Accumulating evidence has shown that mechanical cues and cell-generated traction forces in keratinocytes play an important role in regulating the healing process and wound closure (*Evans et al., 2013*; *Rosińczuk et al., 2016*; *Brugués, 2014*; *Hiroyasu et al., 2016*; *Huang et al., 2017*; *Ladoux and Mège, 2017*). However, the molecular identity of keratinocyte mechanotransducers that control re-epithelialization remains unknown.

**eLife digest** The skin is the largest organ of the body. It enables touch sensation and protects against external insults. Wounding of the skin exposes the body to an increased risk of infection, disease and scar formation. During wound healing, the cells in the topmost layer of the skin, called keratinocytes, move in from the edges of the wound to close the gap. This helps to restore the skin barrier.

Previous research has shown that the mechanical forces experienced by keratinocytes play a role in wound closure. Several proteins, called mechanosensors, perceive these forces and instruct the cells what to do. Until now, it was unclear what kind of mechanosensors control wound healing.

To find out more, Holt et al. studied a recently discovered mechanosensor (for which co-author Ardem Pataputian received the Nobel Prize in 2021), called Piezo1, using genetically engineered mice. The experiments revealed that skin wounds in mice without Piezo1 in their keratinocytes healed faster than mice with normal levels of Piezo1. In contrast, skin wounds of mice with increased levels of Piezo1 in their keratinocytes healed slower than mice with normal levels of Piezo1. The same pattern held true for keratinocytes grown in the laboratory that had been treated with chemicals to increase the activity of Piezo1.

To better understand how Piezo1 slows wound healing, Holt et al. tracked its location inside the keratinocytes. This revealed that the position of Piezo1 changes over time. It builds up near the edge of the wound in some places, and at those regions makes the cells move backwards rather than forwards. In extreme cases, an increased activity of Piezo1 can cause an opening of the wound instead of closing it.

These findings have the potential to guide research into new wound treatments. But first, scientists must confirm that blocking Piezo1 would not cause side effects, like reducing the sensation of touch. Moreover, it would be interesting to see if Piezo1 also plays a role in other important processes, such as development or certain diseases.

Cells are able to sense and detect mechanical forces, converting them into biochemical signals through the process of mechanotransduction. One class of mechanosensors utilized by cells are mechanically activated ion channels which offer the unique ability for cells to rapidly detect and transduce mechanical forces into electrochemical signals (*Nourse and Pathak, 2017*; *Murthy et al., 2017*). The Piezo1 ion channel has been shown to play an important role in a variety of cell types, and it regulates several key biological processes including vascular and lymphatic development, red blood cell volume regulation, stem cell fate, the baroreceptor response, cardiovascular homeostasis, cartilage mechanics, and others (*Li et al., 2014*; *Ranade et al., 2014*; *Pathak et al., 2014*; *Rocio Servin-Vences et al., 2017*; *Zeng et al., 2018*; *Nonomura et al., 2018*; *Cahalan et al., 2015*; *Lee et al., 2014*; *Rode et al., 2017*). Previous studies in MDCK cells and in zebrafish larvae have demonstrated the importance of the channel in homeostatic regulation of epithelial cell numbers (*Gudipaty et al., 2017*; *Eisenhoffer et al., 2012*). As yet, the role of Piezo1 in skin wound healing, an important epithelial function, has not been investigated. We asked whether PIEZO1 may function as a mechanosensor regulating keratinocyte re-epithelialization during the wound healing process. Here, we show that PIEZO1 activity increases cellular retraction, reducing the efficiency of keratinocyte migration and wound healing, and that inhibition of PIEZO1 results in faster wound healing in vitro and in vivo. The channel exhibits dynamic changes in its subcellular localization, concentrating at areas of the wound edge and causing local retraction at these regions.

## Results

### Reduced PIEZO1 accelerates wound healing

Analysis of *Piezo* channel mRNA expression in mouse tissues has previously shown that *Piezo1* is highly expressed in skin, while *Piezo2* is less abundant (*Coste et al., 2010*). To characterize PIEZO1 expression profile in skin, we used a reporter mouse expressing a promoter-less β-geo (β-gal and neomycin phosphotransferase) in-frame with a portion of the PIEZO1 channel (*Ranade et al., 2014*).

LacZ staining of skin tissue from these reporter mice revealed a high expression of PIEZO1 in the epidermal layer of keratinocytes as well as in hair follicles (*Figure 1A*).

Since the global knockout of *Piezo1* is embryonically lethal (*Li et al., 2014*; *Ranade et al., 2014*), we generated an epidermal-specific knockout mouse to investigate whether PIEZO1 plays a role in cutaneous wound healing. The *Krt14^Cre* mouse line was crossed with *Piezo1^fl/fl* mice (*Cahalan et al., 2015*) to generate *Krt14^Cre;Piezo1^fl/fl* mice (hereafter referred to as conditional knockout, cKO) which are viable, develop normally, and feature normal skin sections (*Figure 1—figure supplement 1*), consistent with observations by *Moehring et al., 2020*. qRT-PCR analysis using keratinocytes harvested from *Piezo1*-cKO and littermate control animals confirmed expression of *Piezo1* and not *Piezo2* in control mice (*Figure 1—figure supplement 2*), and showed that *Piezo1* mRNA expression is efficiently abrogated in cells from cKO animals (*Figure 1B*). Furthermore, we also generated a *Piezo1* gain-of-function (GoF) mouse line (*Piezo1*-GoF) which expresses the gain of function (GoF) *Piezo1* mutation, R2482H (*Ma et al., 2018*), in keratinocytes.

To confirm functional change to PIEZO1 in mutant keratinocytes, we performed $Ca^{2+}$ imaging using total internal reflection fluorescence (TIRF) microscopy of keratinocytes isolated from *Piezo1* cKO, GoF, and their respective control (Cre-) littermates. We previously reported that in adherent cells Piezo1 produces $Ca^{2+}$ flickers in response to cell-generated forces in the absence of external mechanical stimulation (*Pathak et al., 2014*; *Ellefsen et al., 2019*). Compared to littermate control (Control$_{cKO}$) cells, keratinocytes from *Piezo1*-cKO mice showed a 63 % reduction in $Ca^{2+}$ flickers, indicating that a majority of $Ca^{2+}$ flickers arise from cell-generated activation of the PIEZO1 channel (*Figure 1C, D, E*, *Figure 1—video 1*). *Piezo1*-GoF cells displayed a nearly threefold increase in the frequency of $Ca^{2+}$ flickers relative to littermate controls (Control$_{GoF}$) (*Figure 1F, G, H. Figure 1—video 2*), further supporting PIEZO1 as a key source of $Ca^{2+}$ flickers. A difference in the frequency of $Ca^{2+}$ flickers between the Control$_{cKO}$ and Control$_{GoF}$ cells was observed, likely arising from different genetic backgrounds of the two strains. For this reason, in all subsequent experiments, mutant keratinocytes are compared to littermate control cells of the same genetic background.

To investigate the function of PIEZO1 in keratinocytes in vivo, we generated full-thickness wounds on the dorsal skin of *Piezo1* cKO, GoF, and their respective control littermates and assessed wound closure (*Figure 1I*). Six days post wounding, *Piezo1*-cKO mice displayed significantly smaller wound areas relative to their control littermates, while *Piezo1*-GoF mice showed larger wound areas, suggesting that increased channel activity leads to impaired rates of wound closure (*Figure 1J*).

To determine whether the effect on wound healing was caused by changes to the rate of keratinocyte re-epithelialization, we mimicked the in vivo wound healing paradigm in vitro. We generated scratch wounds in keratinocyte monolayers to trigger the re-epithelialization process and allowed the monolayers to migrate toward each other (*Figure 1K*). Scratches in monolayers of *Piezo1*-cKO keratinocytes closed faster than those from littermate Control$_{cKO}$ cells (*Figure 1L*, left). Conversely, scratch closure in monolayers of *Piezo1*-GoF keratinocytes was significantly slower (*Figure 1L*, middle). Correspondingly, when the PIEZO1 agonist Yoda1 was added to healing Control$_{cKO}$ monolayers at concentrations greater than 2 µM, scratch wound closure was also significantly impaired (*Figure 1L*, right, *Figure 1—figure supplement 3A*), further supporting PIEZO1 involvement in re-epithelialization. No effect on wound closure was observed when *Piezo1*-cKO monolayers were treated with Yoda1 indicating that inhibition of scratch closure is the result of PIEZO1 activity (*Figure 1—figure supplement 3B*). Collectively, our in vitro and in vivo data demonstrate that the PIEZO1 ion channel plays an important role in wound healing, with *Piezo1* knockout accelerating the healing process (*Figure 1M*).

## PIEZO1 regulates keratinocyte migration

To determine whether the differences in wound closure rates arise due to PIEZO1's effect on keratinocyte motility during the re-epithelialization process, we captured migration dynamics of dissociated single keratinocytes from *Piezo1*-cKO mice. Isolated cells were sparsely seeded onto fibronectin-coated glass-bottom dishes and imaged over several hours using differential interference contrast (DIC) time-lapse imaging (*Figure 2A*, *Figure 2—video 1*). We tracked the position of individual cells in the acquired movies and analyzed the extracted cell migration trajectories using an open-source algorithm, DiPer (*Gorelik and Gautreau, 2014*). The time-lapse images and corresponding cell migration trajectories (*Figure 2B*; *Figure 2—figure supplement 1*) revealed that the migration patterns of *Piezo1*-cKO keratinocytes are distinct from their littermate control cells. To quantify cellular migration,

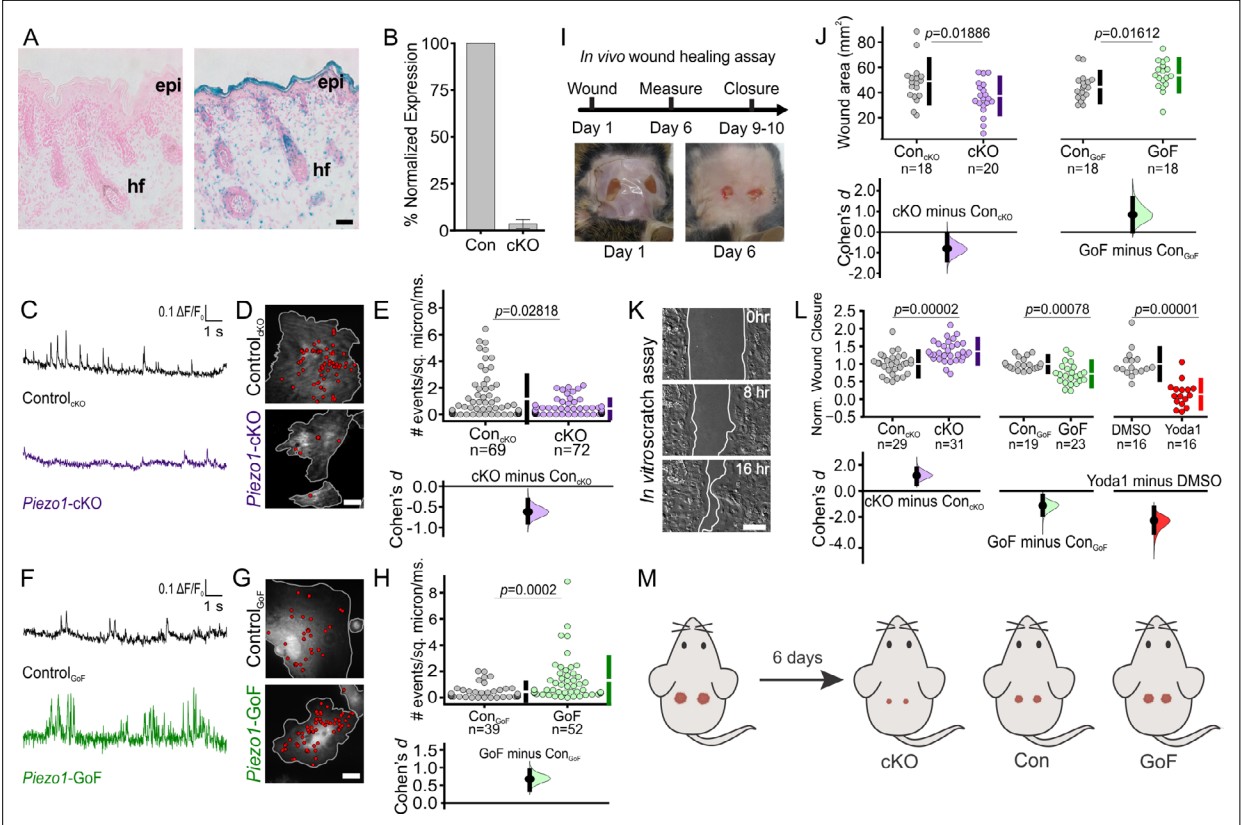

**Figure 1.** PIEZO1 is expressed in keratinocytes, produces Ca²⁺ flickers, and regulates skin wound healing. (**A**) Representative images of LacZ stained Piezo1$^{+/+}$ (*left*) and Piezo$^{+/\beta Geo}$ (*right*) skin sections from P2 (postnatal day 2) mice. Scale bar = 20 µm. epi, epidermis. hf, hair follicle. (**B**) qRT-PCR from primary neonatal keratinocytes of *Piezo1* mRNA expression in Krt14$^{+/+}$ *Piezo1$^{fl/fl}$* (Con) and *Krt14$^{Cre/+}$;Piezo1$^{fl/fl}$* (conditional knockout [cKO]) mice. Data presented as the mean ± SEM. See also ***Figure 1—figure supplements 1–2***. Data collected from two litters. (**C**) Representative examples of Ca²⁺ flickers recorded from Control_cKO (*top*) and *Piezo1*-cKO (bottom) keratinocytes. Traces show fluorescence ratio changes (ΔF/F₀) from a Ca²⁺ flicker site, plotted against time. (**D**) Representative images of sites of Ca²⁺flickers (red dots) overlaid on images of keratinocytes isolated from Control_cKO (*top*) and littermate *Piezo1*-cKO (*bottom*) mice that are loaded with fluorescent Ca²⁺ indicator Cal-520AM. Gray line denotes cell boundary, Scale bar = 20 µm. (**E**) Cumming plot showing frequency of Ca²⁺ flickers in *Piezo1*-cKO and respective Control_cKO cells (*p* value calculated via Mann-Whitney test; Cohen's *d* = −0.6175). n in E denotes the number of videos (i.e., unique fields of view imaged, each of which is composed of one or more cells). Videos were collected from four independent experiments from three litters. See also ***Figure 1—video 1***. (**F, G, H**) Similar to C, D, E but using keratinocytes from Control_GoF (*top*) and *Piezo1*-GoF (*bottom*) mice (*p* value calculated via Mann-Whitney test; Cohen's *d* = 0.6747). Videos were collected from four independent experiments from three litters. See also ***Figure 1—video 2***. (**I**) Diagram of in vivo wound healing model (*top*) and representative wound images at days 1 and 6 (*bottom*). (**J**) Cumming plot showing wound area of *Piezo1*-cKO (*left*) and *Piezo1*-GoF (*right*) groups at day 6 relative to control (Con) littermates (*p* value calculated via two-sample t-test; Cohen's *d* = −0.799 and 0.844, respectively). Control (Con) refers to the Cre-negative littermates in each group. n in J denotes number of wounds measured, with two wounds per animal. (**K**) Representative images of an in vitro scratch assay. White line represents the monolayer edge. Scale bar = 200 µm. (**L**) Cumming plot showing quantification of scratch wound closure in monolayers of keratinocytes isolated from: Control_cKO vs. *Piezo1*-cKO mice (*left; p* value calculated via two-sample t-test; Cohen's *d* = 1.188; images from three independent experiments), Control_GoF vs. *Piezo1*-GoF mice (*middle; p* value calculated via two-sample t-test; Cohen's *d* = −1.128; images from four independent experiments) or DMSO-treated vs 4 µM Yoda1-treated Control_cKO monolayers (*right; p* value calculated via Mann-Whitney test; Cohen's *d* = −2.278; images from three independent experiments). n in L denotes the number of unique fields of view imaged. Data are normalized to the mean scratch closure of the corresponding control condition where one is the average closure distance of the control and 0 is no closure. See also ***Figure 1—figure supplement 3***. (**M**) Schematic illustrating results from in vivo wound healing assay shown in I and J. Mice were wounded (*left*) and after 6 days, wounds of *Piezo1*-cKO mice healed more than Control, whereas wounds from *Piezo1*-GoF mice healed less. Vertical bars in upper Cumming plots denote mean ± s.d.

The online version of this article includes the following video and figure supplement(s) for figure 1:

**Source data 1.** PIEZO1 is expressed in keratinocytes, produces Ca²⁺ flickers, and regulates skin wound healing.

**Figure supplement 1.** *Piezo1*-cKO mice develop normally.

**Figure supplement 2.** *Piezo1* is the primary PIEZO channel expressed in mouse keratinocytes.

**Figure supplement 2—source data 1.** *Piezo1* is the primary PIEZO channel expressed in mouse keratinocytes.

*Figure 1 continued on next page*

*Figure 1 continued*

**Figure supplement 3.** Yoda1 inhibits scratch wound closure in control monolayers but has no effect on closure in *Piezo1*-cKO monolayers.

**Figure supplement 3—source data 1.** Yoda1 inhibits scratch wound closure in Control$_{cKO}$ monolayers but has no effect on closure in *Piezo1*-cKO monolayers.

**Figure 1—video 1.** PIEZO1 Ca$^{2+}$ flickers are reduced in *Piezo1*knockout keratinocytes.

https://elifesciences.org/articles/65415/figures#fig1video1

**Figure 1—video 2.** PIEZO1 Ca$^{2+}$ flickers are increased in *Piezo1* gain-of-function keratinocytes.

https://elifesciences.org/articles/65415/figures#fig1video2

we generated mean squared displacement (MSD) plots which provide a measure of the surface area explored by the cells, and is an indication of the overall efficiency of migration. Interestingly, *Piezo1*-cKO keratinocytes explored a larger area compared to littermate Control$_{cKO}$ cells (*Figure 2C*).

The MSD of a migrating cell is determined by two parameters: directional persistence (propensity of the cell to move in a straight line) and displacement rate (speed). To assess directional persistence, we performed direction autocorrelation analysis, a robust measure of migration directionality that, unlike the more commonly used directionality ratio analysis, is not confounded by differences in migration speed (*Gorelik and Gautreau, 2014*). The direction autocorrelation function for trajectories from *Piezo1*-cKO keratinocytes decayed slower than for littermate Control$_{cKO}$ cells, indicative of fewer turns and a straighter trajectory (*Figure 2D*). The average instantaneous speed calculated by DiPer analysis was higher for *Piezo1*-cKO cells relative to littermate Control$_{cKO}$ cells (*Figure 2E*). Thus, *Piezo1*-cKO keratinocytes migrate significantly faster and straighter. Similarly, we extracted cell migration trajectories from single migrating keratinocytes harvested from *Piezo1*-GoF and littermate Control$_{GoF}$ mice (*Figure 2—figure supplement 2*, *Figure 2—video 2*). We observed that *Piezo1*-GoF keratinocytes also explored a somewhat larger area compared to littermate Con$_{GoF}$ cells, due to the cells migrating straighter with no difference in cell speed (*Figure 2—figure supplement 3*). Overall, our data demonstrate that PIEZO1 regulates keratinocyte migration, with channel knockout resulting in faster migration speed. The effects of the GoF mutation were more complex, and both *Piezo1* knockout and the GoF mutation resulted in straighter trajectories.

## PIEZO1 regulates cell shape and induces a polarized shape

To gain insights into how PIEZO1 may regulate cell migration, we visualized localization of endogenous PIEZO1 in single keratinocytes harvested from a *Piezo1*-tdTomato fusion knock-in reporter mouse (*Ranade et al., 2014*). Using this model, we previously reported punctate membrane localization of endogenous PIEZO1-tdTomato channels in neural stem/progenitor cells and mouse embryonic fibroblasts (*Ellefsen et al., 2019*). Here, we directly imaged endogenous PIEZO1-tdTomato's subcellular localization in individual live, migrating keratinocytes using TIRF imaging and noticed higher PIEZO1 levels at the rear end of the cell (*Figure 3A*, *Figure 3—video 1*). This observation of PIEZO1-tdTomato enrichment at the rear of single migrating cells suggests that PIEZO1 may underlie cell polarization during migration.

To determine whether PIEZO1 may be responsible for generating the polarized shape, we performed cellular morphometrics on the images obtained from the above time-lapse imaging of single cell migration. We used visually aided morpho-phenotyping image recognition (VAMPIRE) (*Phillip et al., 2021*), a high-throughput machine-learning algorithm that analyzes the morphology of individual cells in a population by quantifying shape modes of segmented cells and showing the level of correlation between the shape modes through a dendrogram (*Figure 3B and C*). VAMPIRE classification of the *Piezo1*-cKO and littermate Control$_{cKO}$ keratinocytes into 20 shape modes revealed that *Piezo1*-cKO reduced the proportion of highly polarized shapes and increased the proportion of weakly polarized shapes relative to littermate Control$_{cKO}$ keratinocytes (*Figure 3D and E*). On the other hand, the GoF mutation increased the frequency of polarized and hyper-polarized shapes at the expense of unpolarized or weakly polarized cell shapes (*Figure 3F and G*). Taken together, these results indicate that PIEZO1 activity promotes cell polarization. Based on imaging the localization of endogenous PIEZO1 channels in migrating cells, it appears that this may be mediated by regulation of the channel's subcellular localization.

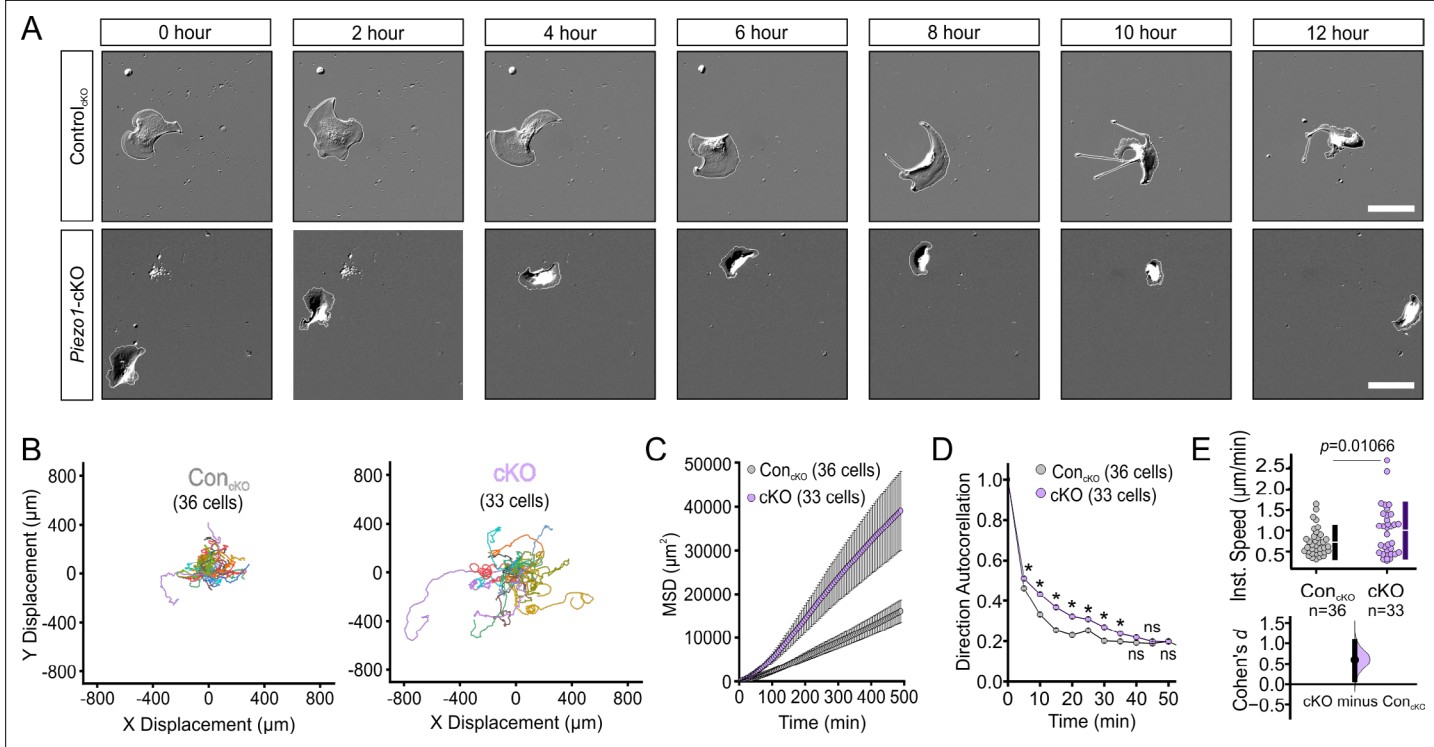

**Figure 2.** PIEZO1 mediates speed and direction during single cell keratinocyte migration. (**A**) Representative differential interference contrast (DIC) images from time-lapse series of individual migrating keratinocytes isolated from Control<sub>cKO</sub> (*top*) and respective *Piezo1*-cKO mice (*bottom*). Thin white lines denote the cell boundary. Scale bar = 25 μm. (**B**) Cell trajectories derived from tracking single keratinocytes during time-lapse experiments. Trajectories are shown with cell position at time point 0 normalized to the origin. See also *Figure 2—figure supplement 1*. (**C**) Mean squared displacement (MSD) analysis of Control<sub>cKO</sub> and *Piezo1*-cKO keratinocytes tracked in B. Average MSD is plotted as a function of time. Error bars (SEM) are smaller than symbols at some points. (**D**) Average direction autocorrelation measurement of *Piezo1*-cKO and Control<sub>cKO</sub> keratinocytes plotted as a function of time interval. * denotes a statistically significant difference, and ns denotes 'not statistically significant'. From left to right: $p$ = 2.0307 × $10^{-4}$, 5.75675 × $10^{-14}$, 3.18447 × $10^{-15}$, 5.34662 × $10^{-10}$, 1.72352 × $10^{-4}$, 1.34648 × $10^{-5}$, 0.01951, 0.13381, 0.61758 as determined by Kruskal-Wallis test. Plotted error bars (SEM) are smaller than symbols. (**E**) Quantitation of the average instantaneous speed from individual *Piezo1*-cKO keratinocytes relative to control cells are shown in a Cumming plot (Cohen's $d$ = 0.6; $p$ value calculated via Kolmogorov-Smirnov test). n in B–E denotes the number of individually migrating cells tracked. See also *Figure 2—figure supplements 2–3* and *Figure 2 videos 1* and *2*. Data are from three independent experiments from two litters. Bars in upper Cumming plots denote mean ± s.d.

The online version of this article includes the following video and figure supplement(s) for figure 2:

**Source data 1.** PIEZO1 mediates speed and direction during single cell keratinocyte migration.

**Figure supplement 1.** *Piezo1*-cKO keratinocytes migrate further.

**Figure supplement 1—source data 1.** *Piezo1*-cKO keratinocytes migrate further.

**Figure supplement 2.** *Piezo1*-GoF keratinocytes migrate straighter.

**Figure supplement 2—source data 1.** *Piezo1*-GoF keratinocytes migrate straighter.

**Figure supplement 3.** Single cell migration of *Piezo1*-GoF keratinocytes.

**Figure supplement 3—source data 1.** Single cell migration of *Piezo1*-GoF keratinocytes.

**Figure 2—video 1.** *Piezo1* knockout affects keratinocyte motility.
https://elifesciences.org/articles/65415/figures#fig2video1

**Figure 2—video 2.** *Piezo1* gain-of-function (GoF) affects keratinocyte speed.
https://elifesciences.org/articles/65415/figures#fig2video2

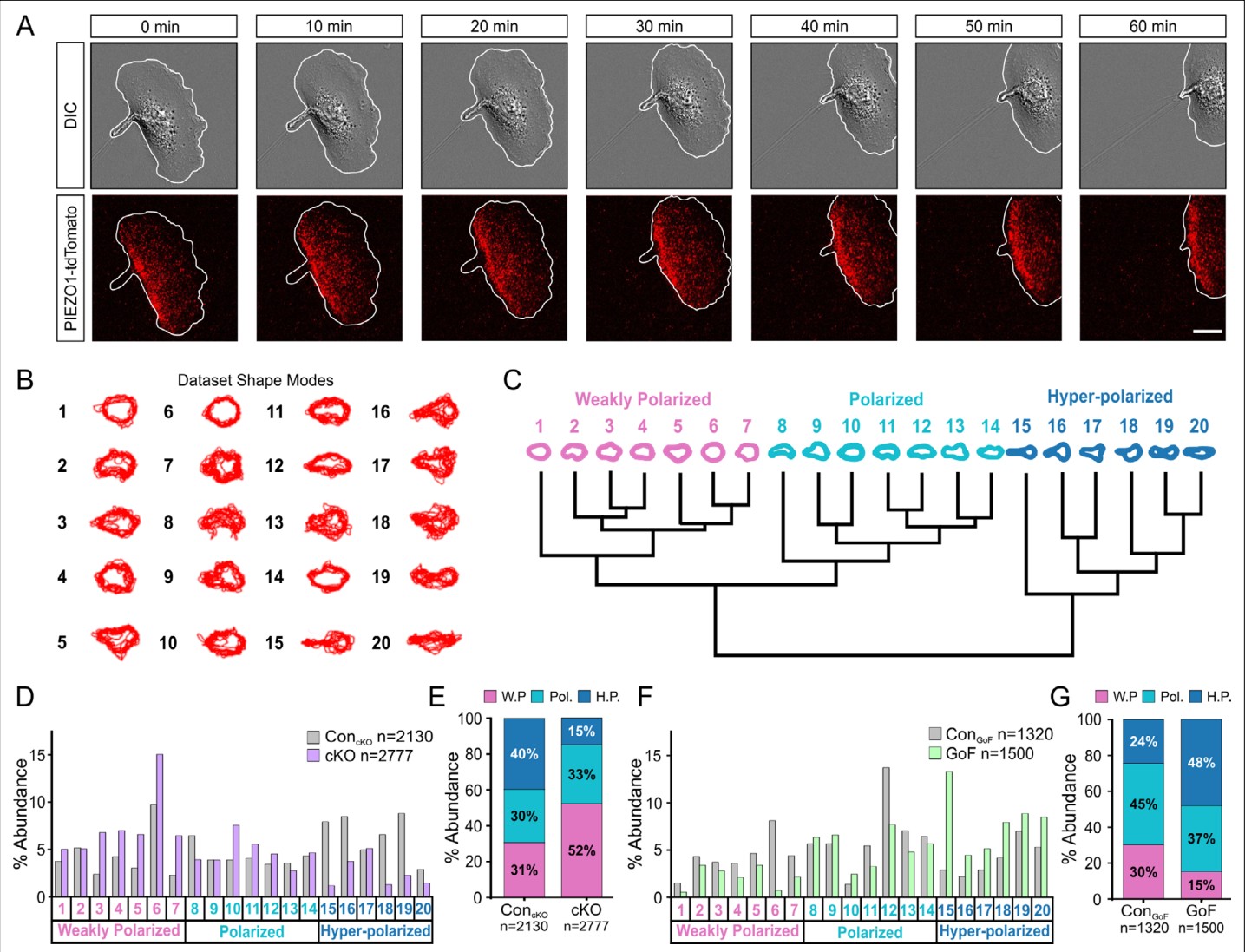

**Figure 3.** PIEZO1 activity promotes cell polarization. (**A**) Time-lapse series of representative differential interference contrast (DIC) (*top*) and total internal reflection fluorescence (TIRF) (*bottom*) images illustrating the location of PIEZO1-tdTomato protein during keratinocyte migration. White lines denote the cell boundary. Scale bar = 15 µm. Images representative of four independent experiments from two litters. See also *Figure 3—video 1*. (**B**) Representative overlays of cell outlines segmented from Control_cKO, *Piezo1*-cKO, Control_GoF, and *Piezo1*-GoF single cell migration time-lapse images and classified into 20 representative shape modes using VAMPIRE (visually aided morpho-phenotyping image recognition), a machine learning algorithm designed for quantification of cellular morphological phenotypes. (**C**) Dendrogram showing the level of correlation between shape modes identified by VAMPIRE. Cell shape modes are classified into the morphologically distinct categories of weakly polarized (*pink*), polarized (*blue*), or hyper-polarized (*dark blue*). (**D**) Bar plots showing the distribution of cell shape modes of Control_cKO (gray; n denotes the number of images analyzed from seven cells and two independent experiments) and *Piezo1*-cKO (purple; n denotes the number of segmented shapes from 12 cells and two independent experiments) cells. (**E**) Stacked bar graphs indicating the proportion of weakly polarized (W.P.) (*pink*), polarized (Pol.) (*blue*), or hyper-polarized (H.P.) (*dark blue*) shape modes in Control_cKO (*left*) and *Piezo1*-cKO (*right*) cells. (**F, G**) Similar to D and E respectively but showing the distribution of cell shape modes for *Piezo1*-GoF (green; n denotes the number of shapes from nine cells and three independent experiments) and respective Control_GoF (gray; n denotes the number of shapes from eight cells and three independent experiments) cells during single cell migration time-lapse experiments. See also *Figure 2 videos 1* and *2*. Bars in D-G denote frequency.

The online version of this article includes the following video and figure supplement(s) for figure 3:

**Source data 1.** PIEZO1 activity promotes cell polarization.

**Figure 3—video 1.** PIEZO1-tdTomato is enriched at the rear of individually migrating keratinocytes.

https://elifesciences.org/articles/65415/figures#fig3video1

## Dynamic PIEZO1 localization informs retraction events to regulate wound healing

We then examined a role for PIEZO1 localization in wounded cell monolayers. We generated a scratch wound in a confluent monolayer of *Piezo1*-tdTomato keratinocytes and imaged spatiotemporal dynamics of PIEZO1-tdTomato localization at the cell-substrate interface using TIRFM imaging together with DIC imaging over a period of several hours. We found that at some regions along the wound margin, PIEZO1-tdTomato was enriched in band-like structures (*Figure 4A*). Interestingly, this enrichment, which was observed a few hours after scratch generation (compare *Figure 4—figure supplement 1* and *Figure 4A*), was also highly dynamic, such that it ebbed and flowed over the course of imaging (*Figure 4—video 1*).

We asked whether regions displaying PIEZO1-tdTomato enrichment migrate differently from regions without enrichment. To systematically assess the relationship between PIEZO1-tdTomato enrichment and wound edge dynamics, we used kymographs to graphically represent PIEZO1-tdTomato position over the imaging period from regions that displayed PIEZO1-tdTomato enrichment at the wound edge (*Figure 4C, E*, *Figure 4—video 1*) and compared them to control fields of view that showed no such channel enrichment throughout the videos (*Figure 4B, D*, *Figure 4—video 2*). PIEZO1-tdTomato enrichment events at the wound edge appeared as linear streaks in the kymographs (*Figure 4E*, left panel) which could be objectively identified by Kymobutler, a deep-learning-based kymograph analysis software (*Jakobs et al., 2019*; *Figure 4E*, middle and right panels). For the kymographs from control fields of view (*Figure 4B*, *Figure 4—video 2*), Kymobutler did not detect any tracks and we did not observe retraction of the wound edge (*Figure 4D*). In fields of view that exhibited PIEZO1-tdTomato puncta enrichment at the wound edge, this channel enrichment was followed by a localized retraction of the wound edge (*Figure 4C*, *Figure 4E*, *Figure 4—video 1*). Kymobutler analysis of PIEZO1-tdTomato tracks overlaid on the DIC kymographs allowed examination of the migration dynamics of the wound edge in relation to Piezo1 enrichment (*Figure 4E*); 72 % of these PIEZO1-tdTomato enrichment tracks displayed a negative slope corresponding to cell edge retraction and aligned with retraction events (*Figure 4G*). In some fields of view, PIEZO1 enrichment and the accompanying retraction lasted for shorter periods of time, and the periods without channel enrichment were accompanied by wound edge protrusion. In other cases, channel enrichment was maintained for several hours and was accompanied by a sustained and overt retraction of the wound edge throughout that period (*Figure 4F*, *Figure 4—video 3*). Thus, enrichment of PIEZO1-tdTomato puncta resulted in wound edge retraction (*Figure 4H*). Importantly, the rear end of single migrating cells, where PIEZO1 is found to localize (*Figure 3A*), is also the site of cellular retraction (*Petrie et al., 2009*; *Yam, 2007*), suggesting a general relationship between PIEZO1 localization and retraction.

## PIEZO1 activation causes cellular retraction

To examine the relationship between PIEZO1 activity and cellular retraction, we examined the effect of chemical activation of PIEZO1 by Yoda1. We first focused on cellular dynamics of single, migrating Control$_{cKO}$ keratinocytes by imaging at a high spatiotemporal resolution. Using DIC imaging, we monitored migrating keratinocytes at 5 second intervals under control conditions and after Yoda1 treatment (*Figure 5A*, *Figure 5—video 1*). Kymographs were used to visualize changes in the cell edge position over time. We observed that under control conditions, the cell edge displayed cycles of protrusion and retraction which was expected since cell migration is known to progress by iterative cycles of protrusion and retraction (*Giannone et al., 2004*; *Giannone et al., 2007*). PIEZO1 activation by 4 µM Yoda1 greatly affected these cycles and resulted in an extremely dynamic cell edge (*Figure 5A*, *Figure 5—video 1*). The frequency as well as the speed of cell edge retractions and protrusions increased upon Yoda1 treatment but resulted in a net cellular retraction over time (*Figure 5A*, *Figure 5—figure supplement 1C*), with some cells demonstrating drastic retraction with Yoda1 treatment (*Figure 5—video 2*). *Piezo1*-cKO keratinocytes did not show an increase in retraction upon treatment with 4 µM Yoda1 (*Figure 5B and* , *Figure 5—video 3*, *Figure 5—figure supplement 1*), demonstrating that Yoda1-induced increase in retraction is mediated by PIEZO1. Additionally, kymographs of *Piezo1*-GoF keratinocytes also showed an increase in cell edge dynamics compared to littermate Control$_{GoF}$ cells further supporting PIEZO1's role in retraction (*Figure 5—figure supplement 1* and *Figure 5—video 4*).

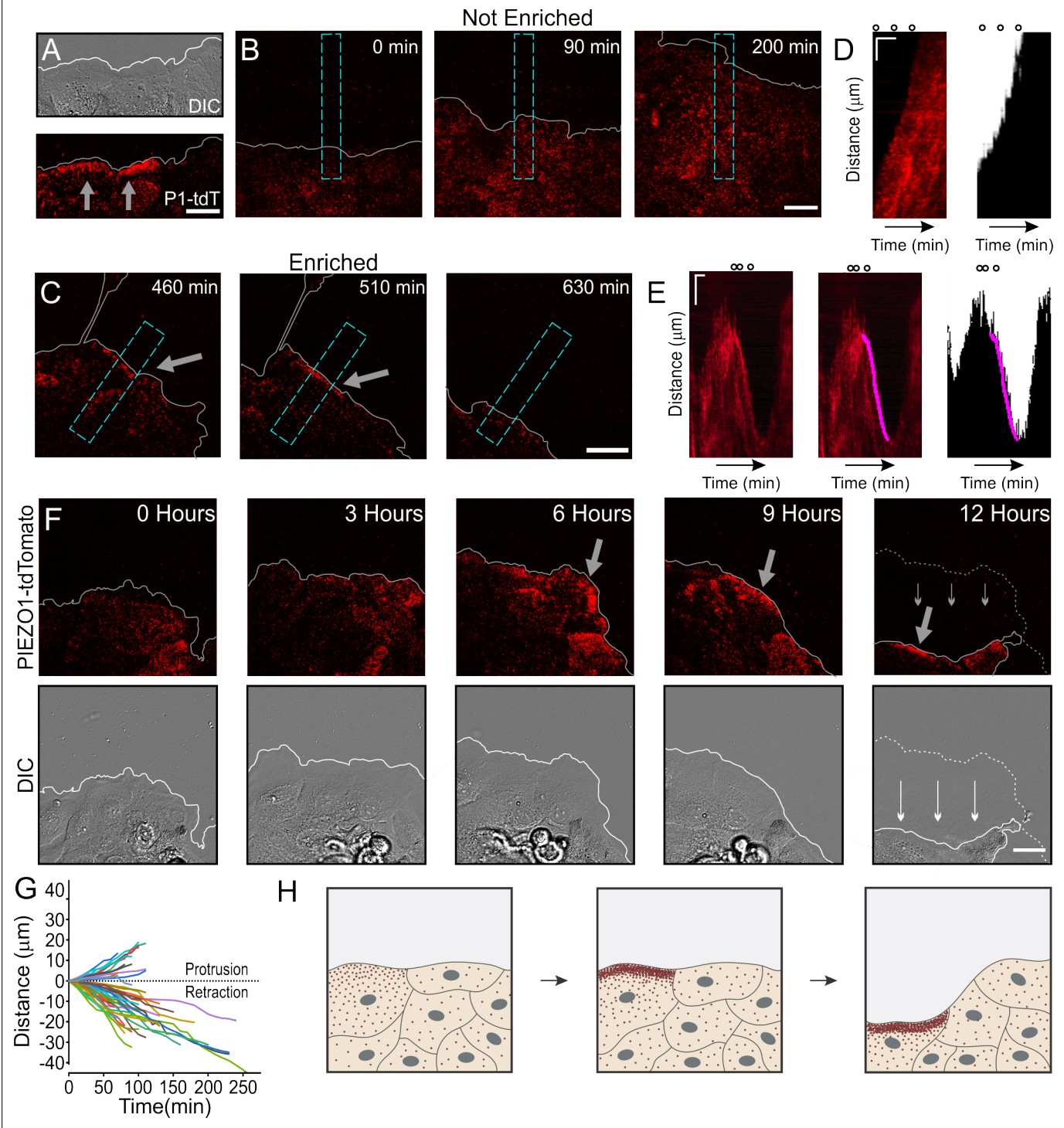

**Figure 4.** Dynamic PIEZO1 channel localization controls wound edge retraction. (**A**) Representative differential interference contrast (DIC) (*top*) and total internal reflection fluorescence (TIRF) (*bottom*) image visualizing the location of PIEZO1-tdTomato (P1-tdT) protein in live, collectively migrating keratinocytes in an in vitro scratch assay. Gray lines denote the boundary of the cell monolayer. Gray arrows indicate regions of PIEZO1 enrichment. Scale bar = 20 μm. See also *Figure 4—figure supplement 1*. (**B, C**) Representative TIRF images taken from time-lapse image series of healing monolayers from PIEZO1-tdTomato keratinocytes highlighting fields of view in which PIEZO1-tdTomato is not enriched (**B**) and enriched (**C**) at the monolayer's leading edge. Gray lines denote the boundary of the cell monolayer. Arrow indicates regions of enrichment. Scale bar = 20 μm. Blue dashed rectangles in D and E depict the regions used to generate kymographs in D and E. TIRF images were acquired every 10 min over a period of 16.6 hr. (**D, E**) *Left panels*: Representative kymographs illustrating PIEZO1-tdTomato puncta dynamics during the time-lapse series shown in B and C, respectively. *Middle panel (for E only):* Magenta line denotes periods of PIEZO1-tdTomato puncta enrichment at the wound edge, identified and

*Figure 4 continued on next page*

*Figure 4 continued*

tracked using the deep learning-based kymograph analysis software, Kymobutler. No Kymobutler tracks were detected in non-enriched regions (**D**). *Right panels*: Representative kymographs from binarized versions of DIC images corresponding to D and E, respectively, with the Kymobutler track output from the middle panel overlaid. The cell monolayer is represented in black, and white denotes cell-free space of the wounded area. Note the PIEZO1-tdTomato enrichment track correlates with periods of cell retraction. Scale bar = 10 µm. Time bar = 100 min. Black open circles on top represent the time-points of images shown in D and E. See also *Figure 4—video 1* and *Figure 4—video 2*. (**F**) DIC (*bottom*) and TIRF (*top*) images during a time-lapse imaging series following scratch generation at 0 hr from a field of view showing sustained PIEZO1-tdTomato localization and marked monolayer retraction. Gray (*top*) and white (*bottom*) lines denote the boundary of the monolayer. Dotted line in the 12 hr image denotes the position of the monolayer at 6 hr; thin arrows indicate direction of monolayer movement during this period. Large gray arrow indicates region of PIEZO1 enrichment. Scale bar = 20 µm. See also *Figure 4—video 3*. (**G**) Plot showing 54 individual PIEZO1-tdTomato Kymobutler tracks from 25 kymographs collected from three independent experiments after normalizing the starting spatial and time coordinates of each track to the origin. (**H**) Schematic of a healing monolayer indicating distributed Piezo1 localization (red dots) following scratch generation (*left*), the development of areas of PIEZO1 enrichment (*middle*), and subsequent retraction of those areas (*right*).

The online version of this article includes the following video and figure supplement(s) for figure 4:

**Source data 1.** Dynamic PIEZO1 channel localization controls wound edge retraction.

**Figure supplement 1.** Absence of PIEZO1-tdTomato enrichment at wound edge immediately after scratch wound generation.

**Figure 4—video 1.** PIEZO1-tdTomato becomes enriched at regions at the edge of keratinocyte monolayers and elicits localized retraction.
https://elifesciences.org/articles/65415/figures#fig4video1

**Figure 4—video 2.** Lack of PIEZO1-tdTomato enrichment in advancing monolayers.
https://elifesciences.org/articles/65415/figures#fig4video2

**Figure 4—video 3.** Persistent PIEZO1-tdTomato enrichment at the wound edge elicits sustained retraction.
https://elifesciences.org/articles/65415/figures#fig4video3

Kymograph-based quantitation is limited to one point along the cell edge. To more objectively investigate the effect that PIEZO1 activation has on cell morphodynamics, cells were segmented for each frame of a DIC time-lapse series for the following conditions: Control$_{cKO}$ cells before and after Yoda1 addition, *Piezo1*-cKO cells, Control$_{GoF}$ cells, and *Piezo*-GoF keratinocytes (*Figure 5C*). By comparing segmented outlines between frames, we could obtain the velocity of the cell edge at every position along the detected boundaries for protrusion events (positive velocities) and retraction events (negative velocities). Yoda1 treatment resulted in a significant increase in edge velocities compared to DMSO-treated Control$_{cKO}$ cells (*Figure 5D*). Yoda1, which is expected to globally activate PIEZO1 channels, resulted in an increase of cell edge velocity during both protrusion and retraction events, though the increase in retraction velocity was greater. In contrast, *Piezo1*-cKO keratinocytes showed a reduction in edge velocities relative to littermate Control$_{cKO}$ cells. Heatmaps of cell edge velocity illustrate the robustness of this response (*Figure 5E*). Consistent with our observations of Yoda1 treatment, *Piezo1*-GoF keratinocytes also showed a significant increase in edge velocities relative to littermate Control$_{GoF}$ (*Figure 5D*, *Figure 5—figure supplement 2*). Interestingly, there was a clear increase in the proportion of retracting positions relative to protruding positions in GoF keratinocytes. These results reveal that PIEZO1 activity regulates cell edge dynamics and further support our observations that PIEZO1 activity increases cellular retraction.

We then asked whether PIEZO1-mediated retraction events observed in single cells are relevant in the context of a wounded cell monolayer. We generated scratch wounds in keratinocyte monolayers and performed DIC time-lapse imaging of the healing monolayer in the presence and absence of 4 µM Yoda1. Monolayers of control keratinocytes advanced forward into the cell-free space under DMSO-treated control conditions (*Figure 5F*, *Figure 5—video 5*), while the presence of 4 µM Yoda1 increased retraction events which prevented the monolayer from advancing far into the wound bed (*Figure 5G*, *Figure 5—video 5*). Remarkably, in 35 % of fields of view monitored in scratch assays we observed that Yoda1 treatment resulted in an increase in scratch area instead of wound closure (*Figure 1L*). In *Piezo1*-cKO monolayers we observed cells protrude forward into the cell-free space to close scratch wounds to a greater extent than controls (*Figure 5H*, *Figure 5—video 6*), while GoF monolayers did so to a lower extent (*Figure 5—figure supplement 3*, *Figure 5—video 7*). Additionally, no effect of Yoda1 addition was seen on the rate of advancement in kymographs taken at the wound edge of *Piezo1*-cKO monolayers (*Figure 5—figure supplement 4*). Collectively, our results show that PIEZO1 activity increases cellular retraction in keratinocytes, both in single cells and monolayers, which has a net effect on cell migration.

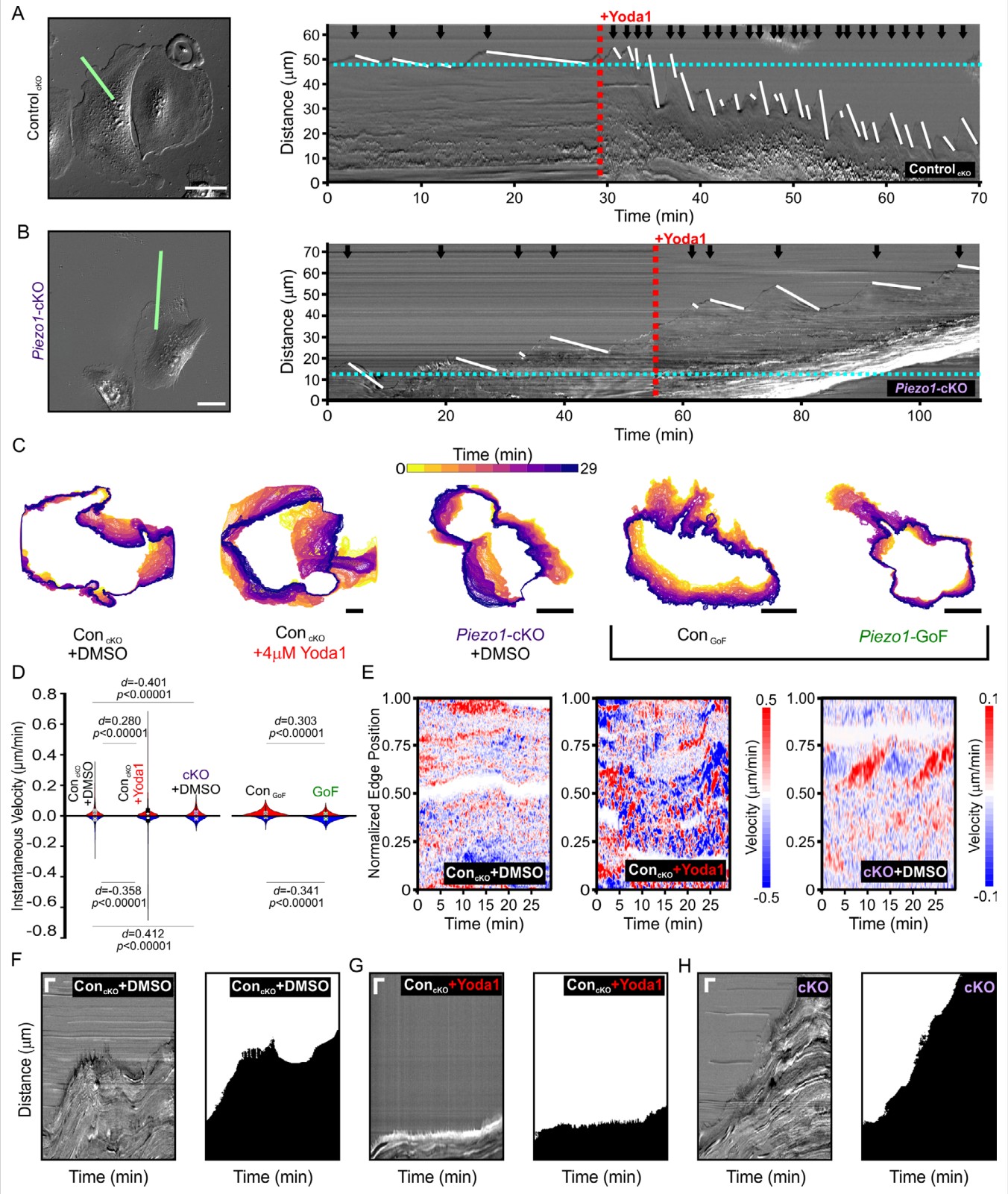

**Figure 5.** PIEZO1 activity promotes cellular retraction and increases edge velocity. (**A**) Image still from a differential interference contrast (DIC) time-lapse series of Control_cKO keratinocytes (*left*). Scale bar = 50 μm. The green line denotes the representative region of interest used for generating kymograph (*right*). The red dotted line denotes the addition of 4 μM Yoda1. The black arrows indicate retraction events, and the slope of the descending white lines denotes the speed of retraction. Dotted blue line denotes starting position of the cell edge. See also ***Figure 5—figure***

*Figure 5 continued on next page*

*Figure 5 continued*

**supplement 1**, **Figure 5—video 1** and **Figure 5—video 2**. Images representative of six independent experiments from cells isolated from three litters. (**B**) Similar to A but using *Piezo1*-cKO keratinocytes with the same annotations. Scale bar = 25 µm. See also **Figure 5—figure supplement 1**, **Figure 5—video 3** and **Figure 5—video 4**. Images representative of seven independent experiments from cells isolated from four litters. (**C**) Representative overlays of cell boundary outlines detected and segmented from DIC time-lapse series of DMSO-treated Control$_{cKO}$, 4 µM Yoda1-treated Control$_{cKO}$, DMSO-treated *Piezo1*-cKO, Control$_{GoF}$, and *Piezo1*-GoF keratinocytes. Brackets separate *Piezo1*-cKO and *Piezo1*-GoF background cell types. Color of cell boundary outline indicates passage of time. Scale bar = 25 µm. (**D**) Violin plots showing the instantaneous protrusion (positive) and retraction (negative) velocities at each position of the segmented cell edges for each frame of time-lapse series videos. Plot shows the combined data from six DMSO and Yoda1 treated Control$_{cKO}$ cells, respectively, nine *Piezo1*-cKO cells, five Control$_{GoF}$, and six *Piezo1*-GoF cells from two independent experiments each. *p* Values calculated using Mann-Whitney test; *d* values show Cohen's *d* which is calculated as test condition minus the respective control condition. (**E**) Representative heatmaps corresponding to the cells shown in C illustrating cell edge velocities along segmented cell boundaries from time-lapse series. See also **Figure 5—figure supplement 2**. (**F, G, H**) Representative DIC kymograph taken at the wound edge and kymograph of the binarized version of the same video, where the cell monolayer is represented in black and cell-free space of the wounded area in white. Kymographs illustrate wound edge dynamics during an in vitro scratch assay performed in (**F**) control (DMSO)-treated monolayers, (**G**) control scratches treated with 4 µM Yoda1, and (**H**) *Piezo1*-cKO monolayers (*right pair*). Scale bar = 20 µm (vertical) and 100 min (horizontal). The data in F, G are representative four independent experiments from keratinocytes from three biological repeats for each condition. The data in H are representative of three independent experiments using keratinocytes from three biological repeats. See also **Figure 5—figure supplements 3–4** and **Figure 5—videos 5–7**.

The online version of this article includes the following video and figure supplement(s) for figure 5:

**Source code 1.** Code used to analyze data shown in **Figure 5**.

**Source data 1.** PIEZO1 activity promotes cellular retraction and increases edge velocity.

**Source data 2.** PIEZO1 activity promotes cellular retraction and increases edge velocity.

**Source data 3.** PIEZO1 activity promotes cellular retraction and increases edge velocity.

**Figure supplement 1.** *Piezo1* gain-of-function (GoF) keratinocytes have increased retraction.

**Figure supplement 1—source data 1.** *Piezo1* gain-of-function keratinocytes have increased retraction.

**Figure supplement 2.** PIEZO1 activity increases edge retraction velocity.

**Figure supplement 2—source data 1.** PIEZO1 activity increases edge retraction velocity.

**Figure supplement 3.** *Piezo1*-GoF monolayers advance slower than controls.

**Figure supplement 4.** Yoda1 shows no effect on *Piezo1*-cKO monolayer advancement.

**Figure 5—video 1.** PIEZO1 agonist Yoda1 increases retraction in keratinocytes.
https://elifesciences.org/articles/65415/figures#fig5video1

**Figure 5—video 2.** PIEZO1 agonist Yoda1 can cause widespread retraction in keratinocytes.
https://elifesciences.org/articles/65415/figures#fig5video2

**Figure 5—video 3.** Yoda1 does not increase retraction in *Piezo1* knockout keratinocytes.
https://elifesciences.org/articles/65415/figures#fig5video3

**Figure 5—video 4.** *Piezo1*-GoF keratinocytes show increased retraction.
https://elifesciences.org/articles/65415/figures#fig5video4

**Figure 5—video 5.** PIEZO1 agonist Yoda1 inhibits migration of keratinocyte monolayers and increases wound edge retraction.
https://elifesciences.org/articles/65415/figures#fig5video5

**Figure 5—video 6.** *Piezo1*-cKO keratinocyte monolayers heal faster than littermate control.
https://elifesciences.org/articles/65415/figures#fig5video6

**Figure 5—video 7.** *Piezo1*-GoF keratinocyte monolayers heal slower than littermate control.
https://elifesciences.org/articles/65415/figures#fig5video7

Taken together, our results demonstrate that PIEZO1 induces cellular retraction to slow single and collective cell migration and thus causes delayed wound healing. We propose that dynamic enrichment of the channel protein serves to locally amplify channel activity and the downstream retraction events. In wound healing monolayers of keratinocytes, PIEZO1 enrichment and the subsequent wound edge retraction provide a molecular mechanism for how PIEZO1 slows wound healing, while absence of the channel accelerates wound healing.

## Discussion

Our findings demonstrate that epidermal-specific *Piezo1* knockout resulted in faster wound healing in mice, and conversely a *Piezo1*-GoF mutation slowed wound healing. We recapitulate this observation in vitro, and show through a combination of orthogonal assays in single cells and in monolayers that PIEZO1 activity modulated by dynamic spatial reorganization of the channel protein slows keratinocyte cell migration during re-epithelialization. These findings provide physiological evidence for the role of a mechanically activated ion channel in wound healing and suggest potential therapeutics through a targeted inhibition of PIEZO1, perhaps applied topically, that could help speed wound healing, potentially reducing risk of infection. However, further in-depth analysis regarding the quality of wound healing following PIEZO1 inhibition is required. Given that speedy wound healing affords an evolutionary advantage and that PIEZO1 activity slows wound healing, a puzzling question arises regarding the role of PIEZO1 expression in keratinocytes. Perhaps, there is an advantage to slower healing in the presence of PIEZO1, or the channel is important for other functions in keratinocytes. Consistent with the latter idea, a recent study by Moehring et al. reports that keratinocyte PIEZO1 is critical for sensory afferent firing and behavioral responses to innocuous and noxious mechanical stimulation (*Moehring et al., 2020*). As such, it would also be important to determine that inhibiting Piezo1 to speed wound healing does not have detrimental effects on normal mechanosensation.

$Ca^{2+}$ signals control many aspects of cell migration, including lamellipodial dynamics, traction force generation, rear retraction, focal adhesion turnover, and migration directionality (*Wei et al., 2012*; *Tsai et al., 2015*; *Canales et al., 2019*). While mechanically activated ion channels were proposed to contribute to $Ca^{2+}$ signaling in single cell migration in vitro as early as 20 years ago (*Lee et al., 1999*; *Doyle et al., 2004*; *Wei et al., 2009*; *Tsai and Meyer, 2012*; *Patkunarajah et al., 2020*), many important questions have remained unanswered, including those related to channel identity, functional effects in collective cell migration, and physiological contribution during wound healing. Using chemical activation as well as genetic modulation of PIEZO1, we provide evidence for its involvement in regulating cellular retraction events in single cells as well as in collective cell migration during keratinocyte re-epithelialization.

Notably, the effects we observed on scratch wound closure and cell edge retraction speeds following Yoda1 activation of PIEZO1 were consistently larger than the effects of *Piezo1*-GoF mutation. This is not surprising as the GoF mutation increases ion flux through PIEZO1 channels, without significantly affecting channel activation (*Glogowska et al., 2017*; *Bae et al., 2013*). Thus, channel activation in GoF keratinocytes is expected to occur based on subcellular localization of PIEZO1 and of the cellular forces that activate them. In contrast, Yoda1 treatment would globally activate PIEZO1 channels in the plasma membrane, leading to a larger effect.

One of the most surprising findings to emerge from our studies is the highly dynamic nature of the spatial localization of PIEZO1 channels in migrating cells. Based on this finding, we propose a novel mechanism regulating cell migration wherein spatiotemporal enrichment of PIEZO1 channels serves to localize and amplify channel activity, and regulate contractile forces, to spatially control cellular retraction events. Piezo1 has been implicated in cell migration in different cell types in vitro (*Maneshi et al., 2018*; *McHugh et al., 2012*; *Chubinskiy-Nadezhdin et al., 2019*; *Hung et al., 2016*; *Li et al., 2015*; *Yu et al., 2020*); however, reports of the effect of the channel on migration have varied in the literature, with channel activity supporting migration in some cell types and inhibiting migration in others. Perhaps, a determining factor of the channel's impact on cell migration is how spatiotemporal localization of the channel is regulated in a given cell type.

Our observations spark several new questions regarding the regulation and functional impacts of PIEZO1's localization and clustering dynamics. Interestingly, molecular dynamics simulations of PIEZO1 suggest that interactions between neighboring PIEZO1 channels may enable cooperative gating between channels (*Jiang et al., 2021*). Clustering of the bacterial mechanosensitive channel MscL has also been reported, but computational simulations predict these clusters in fact decrease channel open probability, providing a defence against unwanted channel gating that may cause osmotic shock (*Paraschiv et al., 2020*; *Grage et al., 2011*). Experimental evidence for functional

interaction between PIEZO1 channels remains pending and two recent patch clamp studies examining this question come to divergent conclusions (*Lewis and Grandl, 2021*; *Wijerathne et al., 2021*).

It is well established that retraction during cell migration occurs due to force generated by myosin II (*Cramer, 2013*; *Ridley, 2011*; *Aguilar-Cuenca et al., 2014*), and we previously showed that myosin II-mediated cellular traction forces elicit localized Piezo1 $Ca^{2+}$ flickers (*Ellefsen et al., 2019*). Since myosin II activation is enhanced by intracellular $Ca^{2+}$ (*Somlyo and Somlyo, 2003*), we speculate that Piezo1 may induce cellular retraction through a feedforward loop between Piezo1 and myosin II: traction force generation by myosin II cause Piezo1-mediated $Ca^{2+}$ influx, which in turn may increase myosin II phosphorylation and force generation through the $Ca^{2+}$-regulated Myosin Light Chain Kinase. Enrichment of Piezo1 in subcellular regions would amplify this effect and result in a localized retraction. Supporting this model, Piezo1-mediated $Ca^{2+}$ events were recently found to elicit retraction of developing endothelial tip cells during vascular pathfinding (*Liu et al., 2020*).

Efficient migration requires protrusion of the cell lamellipodia which is stabilized via the formation of focal adhesions. Without stabilization, lamellipodia protrusions retract backward causing membrane ruffling and reducing migration speed (*Borm et al., 2005*). We observe an increase in cell edge velocity in keratinocytes with increased PIEZO1 activity (in both *Piezo1*-GoF and Yoda1-treated cells) indicating that PIEZO1-mediated effects on cell edge velocity may be the predominant mechanism contributing to inefficient migration. An intriguing paradox observed in our data is that both *Piezo1*-cKO and *Piezo1*-GoF keratinocytes exhibit straighter migration trajectories. We also observe that PIEZO1 localization and increased activity appears linked to cell polarization; thus one possible explanation that warrants further study is that the GoF mutation stabilizes PIEZO1 clusters so that once the channels localize to the rear of the cell, the cell retains its polarization and migration directionality.

Cell migration involves a complex orchestration of events, including sub-cellular dynamics in which cytoskeletal processes in different compartments of the cell need to be implemented in a precise spatiotemporal order. How this is achieved remains an open question. Our findings suggest that spatiotemporal enrichment dynamics of Piezo1 play a role in this coordination. More broadly, our findings provide a mechanism by which nanoscale spatial dynamics of Piezo1 channels can control tissue-scale events, a finding with implications beyond wound healing to processes as diverse as development, homeostasis, disease, and repair.

## Materials and methods

**Key resources table**

| Reagent type (species) or resource | Designation | Source or reference | Identifiers | Additional information |
|---|---|---|---|---|
| Genetic reagent (*mouse*) | *Krt14^{Cre}*;*Piezo1^{fl/fl}* (*Piezo1*-cKO) | This paper | | Generated by breeding *Piezo1^{fl/fl}* mice (Jax stock 029213) with K14^{Cre} mice(The Jackson Laboratory, stock 004782) |
| Genetic reagent (*mouse*) | *Krt14^{Cre}*;*Piezo1^{cx/+}* and Krt14^{Cre};Piezo1^{cx/cx} (*Piezo1*-GoF) | This paper | | Generated by breeding *Piezo1^{cx/cx}* mice (*Ma et al., 2018*) with K14^{Cre} mice. |
| Genetic reagent (*mouse*) | *Piezo1*-tdTomato | JAX; *Ranade et al., 2014* | 029214 (RRID:IMSR_JAX:029214) | |
| Genetic reagent (*mouse*) | *Piezo1* LacZ reporter mice | JAX | 026948 (RRID:IMSR_JAX:026948) | |
| Biological sample (*mouse*) | Murine keratinocytes | UC Irvine, The Scripps Research Institute | | Freshly isolated from P0–P5 mouse pups |
| Antibody | anti-Keratin 14 (Rabbit polyclonal) | Covance | Cat# PRB-155P (RRID:AB_292096) | IF (1:1000) |
| Antibody | anti-Keratin K10 (Rabbit polyclonal) | Covance | Cat# PRB-159P (RRID:AB_291580) | IF (1:1000) |

*Continued on next page*

*Continued*

| Reagent type (species) or resource | Designation | Source or reference | Identifiers | Additional information |
|---|---|---|---|---|
| Antibody | anti-Rabbit Alexa Fluor 488 (Goat polyclonal) | Invitrogen | Cat#A11008 (RRID:AB_143165) | IF (1:1000) |
| Sequence-based reagent | Piezo1 | Thermo Fisher | PCR primers Taqman Assay ID: Mm01241570_g1 | |
| Sequence-based reagent | Piezo2 | Thermo Fisher | PCR primers Taqman Assay ID: Mm01262433_m1 | |
| Sequence-based reagent | Gapdh | Thermo Fisher | PCR primers Taqman Assay ID: Mm99999915_g1 | |
| Sequence-based reagent | Krt14 | Thermo Fisher | PCR primers Taqman Assay ID: Mm00516876_m1 | |
| Commercial assay or kit | SuperScript III | Invitrogen (Thermo Fisher) | Cat#12574026 | Synthesizing cDNA |
| Commercial assay or kit | RNeasy kit | Qiagen | | Isolating RNA |
| Chemical compound, drug | Cal-520 AM | AAT Bioquest Inc | Cat#21130 | |
| Chemical compound, drug | Yoda1 | TOCRIS | 558610 | |
| Software, algorithm | Origin Pro | Originlab | OriginPro (RRID:SCR_014212) | |
| Software, algorithm | VAMPIRE | *Phillip et al., 2021* | Version 1.0 (RRID: SCR_021721) | https://github.com/kukionfr/VAMPIRE_open |
| Software, algorithm | ADAPT | *Barry et al., 2015* | (RRID: SCR_006769) | https://github.com/djpbarry/Adapt |
| Software, algorithm | DiPER | *Gorelik and Gautreau, 2014* | (RRID:SCR_021720) | |
| Software, algorithm | Flika | *Ellefsen et al., 2019* | Version 0.2.17 (RRID: SCR_021719) | https://flika-org.github.io/ https://github.com/kyleellefsen/detect_puffs |
| Software, algorithm | Cell Tracker | *Piccinini et al., 2016* | Version 1.1 (RRID:SCR_021718) | http://celltracker.website/index.html |
| Software, algorithm | Cellpose | *Stringer et al., 2021* | Version 0.06 (RRID:SCR_021716) | https://github.com/MouseLand/cellpose |
| Software, algorithm | ilastik | *Berg et al., 2019* | Version 1.4.0b13 (RRID:SCR_015246) | https://www.ilastik.org/ |
| Software, algorithm | Wolfram Mathematica | Wolfram | Wolfram Mathematica 12 | |
| Software, algorithm | FIJI | *Schindelin et al., 2012* | Version 1.53 c (RRID:SCR_002285) | https://imagej.net/software/fiji/ |
| Software, algorithm | Python | https://www.python.org/ | Version 3.7.0 (RRID:SCR_008394) | |
| Software, algorithm | Kymobutler | *Jakobs et al., 2019* | Version: V1v0v2.wl (RRID:SCR_021717) | https://github.com/alexlib/KymoButler-1 |
| Other | DAPI stain | Invitrogen | D1306 (RRID:AB_2629482) | 1:50,000 |

## Animals

All studies were approved by the Institutional Animal Care and Use Committee of University of California at Irvine and The Scripps Research Institute, as appropriate, and performed in accordance with their guidelines. *Piezo1* LacZ reporter mice (JAX stock 026948) and *Piezo1*-tdTomato reporter mice, expressing a C-terminal fusion of *Piezo1* with tdTomato (*Piezo1*-tdTomato; JAX stock 029214), were

generated in a previous study (*Ranade et al., 2014*). Skin-specific *Piezo1*-cKO mice were generated by breeding *Piezo1*$^{fl/fl}$ mice (*Cahalan et al., 2015*) (Jax stock 029213) with K14$^{Cre}$ (The Jackson Laboratory, stock 004782). Skin-specific *Piezo1*-GoF mice were generated by breeding mice with conditional GoF *Piezo1* allele (*Piezo1*$^{cx/cx}$ mice [*Ma et al., 2018*]) with K14$^{Cre}$ mice. *Piezo1*$^{fl/fl}$ mice were generated in C57BL/6 background and *Piezo1*$^{cx/cx}$ mice were initially generated in BALB/c background and then maintained in C57BL/6 for >10 generations. K14$^{Cre}$ mice were in the C57BL/6 background.

## Keratinocyte isolation

P0–P5 mice were anesthetized with ice prior to decapitation. Bodies were placed in 10 % povidone for 1 min, rinsed with sterile PBS, prior to soaking in 70 % ethanol for a further minute, and rinsed again with sterile PBS. Subsequently, the entire upper dorsal skin above the abdomen was separated from the body. Dorsal skin was left to dissociate in either 0.25 % trypsin/EDTA (Gibco) for 1 hr at 37°C or 1 × dispase solution (CellnTec CnT-DNP-10) at 4 °C overnight for 15–18 hr. After incubation, the epidermis was gently separated from the dermis, laid flat, dorsal side down in Accutase (CellnTec CnT-Accutase-100) and incubated for 30 min at room temperature. The epidermis was then transferred to a dish of either CnT-02 or CnT-Pr media (CellnTec), supplemented with 10 % FBS and 1 % penicillin/streptomycin. The epidermis was cut into small pieces with scissors prior to agitation on a stir plate for 30 min. Cells were then filtered through a 70 µm cell strainer (Falcon) and spun down at 1200 rpm for 5 min. The pellet was resuspended in CnT-Pr media (CellnTec) supplemented with ISO-50 (1:1000) (CellnTec) and Gentamicin (50 µg/ml) (Thermo Fisher), prior to counting and plating.

## Keratinocyte culture

For live-cell imaging, primary keratinocytes were plated on #1.5 glass-bottom dishes (Mat-Tek Corporation) coated with 10 µg/ml fibronectin (Fisher Scientific, CB-40008A). For monolayer experiments, cells were plated at $1.5 \times 10^5$ cells/dish in the 14 mm glass region of dishes. For sparse cell migration experiments and for Ca$^{2+}$ imaging experiments, cells were plated at $1.5 \times 10^4$ cells/dish in the 14 mm glass region of dishes. Keratinocytes were imaged following at least 2 days in Cnt-Pr-D (CellnTec) differentiation media.

## Real-time quantitative PCR

After initial keratinocyte isolation and filtering through a 70 µm cell strainer, cells were filtered again through a 40 µm strainer. The filtered solution was spun down and a cell pellet was obtained for RNA isolation. Total RNA was isolated using the RNeasy kit (Qiagen), following which cDNA was synthesized using Superscript III (Invitrogen) and was used for subsequent qPCR experiments (ABI 7900HT fast real-time system). qPCR Taqman probes (Thermo Fisher) used were *Piezo1*: Assay ID Mm01241570_g1; *Piezo2*; Assay ID Mm01262433_m1, *Krt14*; Assay ID Mm00516876_m1 and *Gapdh*; Assay ID Mm99999915_g1. PCR reactions were run in triplicate.

## X-Gal/LacZ staining

Dorsal skin was harvested as described above, cryopreserved in OCT, and sectioned into 8 µm thick slices. Skin cryosections were allowed to completely dry prior to being fixed in 'fix buffer' composed of 1 × PBS, 5 mM EGTA (Sigma Cat#E4378), 2 mM MgCl$_2$, 0.2 % glutaraldehyde (Sigma Cat#G-7776), pH 7.4 for 15 min at room temperature. Next they were washed with 'wash buffer' composed of 1 × PBS, 2 mM MgCl$_2$ twice for 5 min each. X-gal staining buffer composed of 1 × PBS, 2 mM MgCl$_2$, 5 mM potassium ferrocyanide [K$_4$Fe(CN)$_6$·3H$_2$0] (Sigma Cat# P-9287), 5 mM potassium ferricyanide [K$_3$Fe(CN)$_6$] (Sigma Cat#P-8131), and 1 mg/ml X-gal [5-bromo-4-chloro-3-indolyl-β-D-galactoside] was made fresh. Then, tissue slides were incubated overnight at 37 °C in the 'X-gal staining buffer' inside a humidified chamber. The following day, slides were rinsed with 1 × PBS and counterstained with Nuclear Fast Red. Slides were then fixed with 4 % PFA for longer preservation.

## Immunofluorescence staining

For immunostaining of skin sections in *Figure 1—figure supplement 1*, dorsal skin was prepared and sectioned as for X-Gal staining. Skin cryosections were fixed for 10 min in cold acetone, washed twice in 1 × PBS prior to blocking for 30 min in 10 % normal goat serum at room temperature. Primary antibodies used were Rabbit anti-Keratin 14 (Covance, Cat#PRB-155P), 1:1000 (1 µg/ml), and Rabbit

anti-Keratin 10 (Covance, Cat#PRB-159P), 1:1000 (1 µg/ml). Secondary antibody used was Goat anti-Rabbit Alexa Fluor 488 (Invitrogen, Cat#A11008), 1:1000. Nuclei were stained by DAPI (Invitrogen, Cat#D1306), 1:50,000. All antibody incubations were performed at room temperature, for 1 hr in 1 % BSA in PBS. Slides were mounted in gelvatol containing DAPI.

## Microscopy and image analysis
### Microscopy
Unless otherwise stated, in vitro images were taken using an Olympus IX83-ZDC microscope, equipped with an automated four-line cellTIRF illuminator. A full enclosure environmental chamber (Tokai Hit) allowed cells to be imaged at 37 °C with 5 % $CO_2$ ensuring optimal cell health during time-lapse experiments. Stage movement was controlled by a programmable motorized stage (ASI) while an Olympus ZDC autofocus control unit allowed for samples to remain in focus throughout imaging periods. The open-source microscopy software µManager was used to control the microscope and acquire images for all except *Figures 1A, I and 5A* and *Figure 1—figure supplement 1*. Images for *Figures 1D,G, 4 and 5*, *Figure 3A*, *Figure 4*, *Figure 5B*, *Figure 4—figure supplement 1*, *Figure 5—figure supplement 1*, and *Figure 1—videos 1* and *2*, *Figure 3—video 1*, *Figure 4—videos 1–3*, *Figure 5—video 2*; *Figure 5—video 3*; *Figure 5—video 4* were taken using a PLAPO 60 × oil immersion objective with a numerical aperture of 1.45. Images for *Figure 5A*, *Figure 5—video 1* were taken using a UPlanSApo 40 × dry objective with a numerical aperture of 0.95. Images taken for *Figure 1K*, *Figure 2A*, *Figure 3*, *Figure 5F, G, H*, *Figure 5—figure supplement 3*, and *Figure 2—videos 1* and *2*, *Figure 5—video 5* were taken using a UPlanSApo 10 × dry objective with a numerical aperture of 0.40. Images for *Figures 1D, G, K, 2 and 5*, all figure supplements except *Figure 1—figure supplement 1* and all videos were acquired using a Hamamatsu Flash 4.0 v2+ scientific CMOS camera. Images for *Figure 1—figure supplement 1B* were taken using a Hamamatsu C4742-95-12ER Digital CCD camera.

### Imaging Piezo1 $Ca^{2+}$ flickers (Figure 1C-H, Figure 1—videos 1 and 2)
As described previously (*Ellefsen et al., 2019*), TIRF microscopy was used for the detection of $Ca^{2+}$ flickers. Keratinocytes were loaded through the incubation of 2 µM Cal-520 AM (AAT Bioquest Inc) with 0.04 % Pluronic F-127 (Thermo Fisher) in phenol red-free DMEM-F12 (Cat#11039047, Gibco) for 30–35 min at 37 °C, washed three times, and incubated at room temperature for 10–15 min prior to imaging. Cells were imaged at room temperature in a bath solution comprising 148 mM NaCl, 3 mM KCl, 3 mM $CaCl_2$, 2 mM $MgCl_2$, 8 mM glucose, and 10 mM HEPES (pH adjusted to 7.3 with NaOH, osmolarity adjusted to 313 mOsm/kg with sucrose). Cal-520 fluorescence was elicited by excitation with a 488 nm laser line and images were acquired at a frame rate of 9.54 frames/s using a Hamamatsu Flash 4.0 v2+ scientific CMOS camera.

### Piezo1 $Ca^{2+}$ flicker analysis (Figure 1C-H, Figure 1—videos 1 and 2)
$Ca^{2+}$ flickers were automatically detected as previously described (*Ellefsen et al., 2019*) using the detect_puffs plugin (https://github.com/kyleellefsen/detect_puffs) for the open-source image processing program, Flika (https://flika-org.github.io/). This plugin was used to identify and localize flicker events in recorded videos. Each video is a microscope field of view which contains one or more keratinocytes. To normalize any potential variability in cell number or size between samples, flicker frequency by cell area was computed for each field of view. Cell area was measured by using the FIJI (*Schindelin et al., 2012*) plugin SIOX: Simple Interactive Object Extraction to create binary masks and compute cell area.

### In vitro wound healing assay (Figure 1L, Figure 1—figure supplement 3)
Primary keratinocytes were either densely plated onto Mat-Tek dishes or seeded into a two-well silicone insert in a 35 mm dish (ibidi, 81176). Cells were cultured until monolayer confluence was reached. Subsequently, scratch wounds were generated by either scratching monolayers with a 10 µl pipette (for cells plated on Mat-Tek dishes) or by removing the barrier insert from the dish to create a 500 µm cell-free gap (for cells in ibidi dishes). The dishes were washed with cell culture medium to remove floating cells and cell debris. For pharmacology experiments, Yoda1 or equivalent concentration of

DMSO control was added into the dishes immediately prior to imaging. Dishes were imaged by either DIC or phase-contrast microscopy at 37 °C (5 % $CO_2$) for indicated time points.

## Single cell tracking assay (Figure 2, Figure 2—videos 1 and 2; Figure 2—figure supplements 1–3)

Primary keratinocytes sparsely seeded on fibronectin-coated glass-bottom dishes were allowed to migrate freely for up to 16.67 hr at 37 °C with 5 % $CO_2$ in bath solution composed of Cnt-Pr-D (CellnTec) culture media with extracellular $Ca^{2+}$ concentration adjusted to 1.2 mM. Time-lapse DIC images at multiple microscope fields of view were acquired in each dish of cells at 5 min intervals for the imaging period. When collecting trajectories we only considered cells which (1) stay within the field of view during the imaging period and (2) did not come into contact with other keratinocytes. The center of the cell body was the tracked position of the cell. The initial positions of cells were manually identified, after which the positions of migrating cells were automatically tracked using the Cell Tracker software (https://celltracker.website/index.html) (*Piccinini et al., 2016*). Cell trajectories were logged and exported into Microsoft Excel. Further analysis was subsequently performed using the published open-source algorithm, DiPer (*Gorelik and Gautreau, 2014*) to obtain average instantaneous speed, MSD, directionality analysis, and trajectory flower plots.

## VAMPIRE shape mode analysis (Figure 3B-G)

For analyses of cellular morphometrics, time-lapse images from single cell tracking assays were manually segmented using the generalist deep learning-based segmentation algorithm, Cellpose (https://github.com/MouseLand/cellpose) (*Stringer et al., 2021*). Segmented outputs from Cellpose were manually refined using FIJI to ensure the accuracy of detected shapes before being fed into the VAMPIRE algorithm (*Phillip et al., 2021*) (https://github.com/kukionfr/VAMPIRE_open). VAMPIRE allows the profiling and classification of cells into shape modes based on equidistant points along cell contours. The number of coordinates to extract cell contours was set to 400 to ensure accurate representation of cell shapes and 20 shape modes were used for the dataset.

## PIEZO1-tdTomato time-lapse imaging (Figures 3A and 4, Figure 4—figure supplement 1, Figure 3—video 1, Figure 4—videos 1–3)

Primary PIEZO1-tdTomato keratinocytes were cultured either sparsely or as confluent monolayers. Monolayers were scratched using a 10 µl pipette tip immediately prior to imaging. Experiments were performed in Cnt-Pr-D (CellnTec) culture media with added 1.2 mM $Ca^{2+}$. Either single cells or, for monolayer scratch experiments, regions along the initial wound edge were marked using a programmable stage and imaged throughout the imaging period as cells migrated to close the wound. PIEZO1-tdTomato channels were illuminated using a 561 nm laser and imaged using TIRF microscopy. TIRF and DIC snapshots at regions of interest were sequentially acquired every 10 (*Figures 3A and 5B* and *Figure 3—video 1*, *Figure 4—video 1*) or 30 min (*Figure 5A, C and F*, *Figure 4—figure supplement 1*, *Figure 4—videos 2* and *3*) over the course of up to 16.67 hr. To increase the signal-to-background ratio, time-lapse images were processed from the original recording by subtracting every frame of a movie by (1) the median value z-projection image and then (2) the minimum value z-projection image. The FIJI (*Schindelin et al., 2012*) plugin KymoResliceWide was then used to construct kymographs by taking the average intensity of the transverse of a 100 pixel wide line drawn at ROIs along the wound edge. ROIs were chosen so that they would capture the wound edge for the entire duration it was present within the microscope field of view. At least one kymograph was generated from each unique microscope field of view. Impartial identification and tracking of periods of channel enrichment was performed using the Kymobutler tool in Wolfram Mathematica (https://gitlab.com/deepmirror/kymobutler) (*Jakobs et al., 2019*). Only puncta located at the wound edge that were successfully tracked for at least 70 min were collected for further analyses. The pixel classification function of the computer vision software, Ilastik (https://www.ilastik.org/) (*Berg et al., 2019*), was used to classify DIC images and create binarized movies used for generating binarized kymographs in *Figure 5D and E*.

### Migration dynamics assay (Figure 5A-B, Figure 5—figure supplement 1A; Figure 5—videos 1–4)

Cells were imaged via DIC microscopy at 37 °C with 5 % $CO_2$, with snapshots taken at 5 s intervals for at least 30 min in Cnt-Pr-D culture media (CellnTec) with added 1.2 mM $Ca^{2+}$ and 0.0004 % DMSO. After acquiring baseline images, the media was changed and 4 µM Yoda1 was added to the bath solution. After allowing the drug to act for 5–7 min, imaging was resumed for at least 55 min. Experiments were performed multiple times on independent experiment days. Kymographs were created from representative cells using the FIJI (*Schindelin et al., 2012*) plugin KymoResliceWide (https://imagej.net/KymoResliceWide). Kymographs were built by taking one pixel width lines at ROIs along the cell's leading edge.

### ADAPT analysis of keratinocyte morphodynamics (Figure 5C-E, Figure 5— figure supplement Figure 5—figure supplements 1C and 2)

DIC time-lapse images from migration dynamics time-lapse series were binarized using the pixel classification function of the computer vision software, Ilastik (https://www.ilastik.org/) (*Berg et al., 2019*). Binarized images were then analyzed using the FIJI (*Schindelin et al., 2012*) plugin ADAPT (https://github.com/djpbarry/Adapt) (*Barry et al., 2015*) in order to create cell boundary outline overlays and determine the velocity at each position along the cell edge. A Python script used to transform velocity map ADAPT outputs into matrices which were then used to generate heatmaps and velocity violin plots.

### Wound edge dynamics assay (Figure 5F-H, Figure 5—figure supplements 3 and 4, Figure 5—videos 5–7)

Similar to the in vitro wound healing assay described above, primary keratinocytes were densely seeded and cultured to form monolayers. Scratch wounds were generated by scratching monolayers with a 10 µl pipette tip immediately prior to imaging. Dishes were washed 3 × with cell culture media to remove cell debris. Yoda1 (4 µM) or equivalent concentration of DMSO control was added to the bath media immediately prior to imaging. DIC snapshots were taken every 5 min at ROIs along the wound edge for at least 9 hr. Representative kymographs were created using one pixel-wide ROIs using KymoResliceWide. Binarized movies generated by using the FIJI (*Schindelin et al., 2012*) plugin Trainable Weka Segmentation (*Arganda-Carreras et al., 2017*) (https://imagej.net/Trainable_Weka_Segmentation) were used to create binarized kymographs.

## In vivo wound healing assay

Adult (3–4 months) male and female mice were anesthetized with isoflurane and placed on a heated blanket. The dorsal hair was shaved and further removed by hair-removal cream. Two full-thickness wounds were created in the upper dorsal skin above the abdomen using a 4 mm wide dermal biopsy punch (Integra LifeSciences Corporation). Wounded areas were patched with medical dressing, Tegaderm (3 M). Wound sizes were measured with a scale loupe (Peak Optics, #1975) at day 6 to compare healing progress. Both the short (dS) and long (dL) diameters of the oval-shaped wounds were measured and used to calculate an overall wound area using the equation: dS × dL × π.

## Statistical analysis

Sample sizes are indicated in corresponding figures. Cumming estimation plots were generated and Cohen's *d* in all plots (except *Figure 5D*) was calculated using an online estimation stats tool (https://www.estimationstats.com) (*Ho et al., 2019*). Estimation plots show the raw data plotted on the upper axes with bars beside each group denoting the sample mean ± s.d.; the mean difference and Cohen's *d* effect size is plotted on the lower axes. On the lower plot, the mean difference is depicted as a dot; the 95 % confidence interval is indicated by the ends of the bold vertical error bar. OriginPro 2020 (OriginLab Corporation) was used for calculating *p* values (for all figures except for *Figure 5D*) and generating plots used in *Figure 1B*, *Figure 2B, C, D*, *Figure 3D, E, F, G*, *Figure 4G*, *Figure 5D*, *Figure 1—figure supplement 2*, and *Figure 2—figure supplements 1–3*. All plots generated in OriginLab are presented as the mean ± SEM. Statistical tests used to calculate *p* values are indicated in figure legends. For *Figure 5D*, a Python script was used for calculating Cohen's *d* and *p* values.

## Data availability

The datasets plotted in *Figure 1E,H,J,L*, *Figure 2B, C, D, E*, *Figure 3D, E, F, G*, *Figure 4G*, *Figure 5E*, *Figure 1—figure supplements 2–3*, *Figure 2—figure supplements 1–3*, and *Figure 5—figure supplements 1C and 2* have been uploaded as source data files. Source data files for *Figure 5C and D* have been uploaded to Dryad (doi:10.5061/dryad.hdr7sqvjr).

## Code availability

Code used to analyze ADAPT heatmaps and provide statistical analysis for velocity violin plots shown in *Figure 5D* has been supplied as a supplementary datafile. This code has also been made publicly available as a jupyter notebook uploaded to Github.

## Acknowledgements

We thank Dr. Jamison Nourse and Gabriella Bertaccini for technical support, Vivian Leung for help with illustrations, Dr. Ghaidaa Kashgari for helpful discussions, and members of the lab for comments on the manuscript. This work was supported by NIH grants DP2AT010376, R01NS109810 and a seed grant funded through 5P30AR075047-03 to MMP; NIH grants R01HL143297 and R01AR051219 to AP, a James H Gilliam Fellowship for Advanced Study (GT11549) from the Howard Hughes Medical Institute to MMP and JRH, a postdoctoral fellowship from the George Hewitt Foundation for Medical Research to W-ZZ, and a seed grant to JRH and W-ZZ from the UCI NSF-Simons Center for Multiscale Cell Fate Research (funded by NSF grant DMS1763272 and a Simons Foundation grant 594598). AP is an investigator of the Howard Hughes Medical Institute.

## Additional information

### Funding

| Funder | Grant reference number | Author |
| --- | --- | --- |
| National Institutes of Health | DP2AT010376 | Medha M Pathak |
| National Institutes of Health | R01NS109810 | Medha M Pathak |
| National Institutes of Health | R01HL143297 | Ardem Patapoutian |
| Howard Hughes Medical Institute | GT11549 | Jesse R Holt<br>Medha M Pathak |
| George Hewitt Foundation for Medical Research | | Wei-Zheng Zeng |
| National Science Foundation | DMS1763272 | Jesse R Holt<br>Wei-Zheng Zeng |
| Simons Foundation | 594598 | Jesse R Holt<br>Wei-Zheng Zeng |
| Howard Hughes Medical Institute | | Ardem Patapoutian |
| National Institutes of Health | R01AR051219 | Ardem Patapoutian |
| National Institutes of Health | 5P30AR075047-03 | Medha M Pathak |

The funders had no role in study design, data collection and interpretation, or the decision to submit the work for publication.

## Author contributions
Jesse R Holt, Methodology, Methodology, Software, Validation, Visualization, Writing – original draft, Data curation; Wei-Zheng Zeng, Data curation, Formal analysis, Methodology, Methodology, Conceptualization, Validation, Data curation; Elizabeth L Evans, Data curation, Formal analysis, Methodology, Validation, Visualization, Writing – original draft, Data curation; Seung-Hyun Woo, Funding acquisition, Data curation, Formal analysis, Methodology, Methodology, Conceptualization, Validation, Data curation; Shang Ma, Data curation, Formal analysis, Methodology, Conceptualization, Data curation; Hamid Abuwarda, Formal analysis, Methodology, Data curation; Meaghan Loud, Investigation; Ardem Patapoutian, Conceptualization, Funding acquisition, Project administration, Resources, Supervision, Writing – review and editing; Medha M Pathak, Conceptualization, Data curation, Formal analysis, Funding acquisition, Methodology, Methodology, Funding acquisition, Conceptualization, Resources, Validation, Visualization, Writing – original draft, Data curation

## Author ORCIDs
Jesse R Holt http://orcid.org/0000-0001-8136-6394
Elizabeth L Evans http://orcid.org/0000-0003-1237-7262
Ardem Patapoutian http://orcid.org/0000-0003-0726-7034
Medha M Pathak http://orcid.org/0000-0002-6518-3085

## Ethics
All studies were approved by the Institutional Animal Care and Use Committee of University of California at Irvine (protocol number AUP-19-184) and The Scripps Research Institute (protocol number 08-0136-4), as appropriate, and performed in accordance with their guidelines.

## Decision letter and Author response
Decision letter https://doi.org/10.7554/eLife.65415.sa1
Author response https://doi.org/10.7554/eLife.65415.sa2

# Additional files

## Supplementary files
• Transparent reporting form

## Data availability
The datasets for graphs included in each figure have been made available as source data files. Source data files for Figure 5C and 5D have been uploaded to Dryad (https://doi.org/10.5061/dryad.hdr7sqvjr).

The following dataset was generated:

| Author(s) | Year | Dataset title | Dataset URL | Database and Identifier |
| --- | --- | --- | --- | --- |
| Holt JR, Zeng WZ, Evans EL, Woo SH, Ma S, Abuwarda H, Loud M, Patapoutian A, Pathak MM | 2021 | Spatiotemporal dynamics of PIEZO1 localization controls keratinocyte migration during wound healing | https://doi.org/10.5061/dryad.hdr7sqvjr | Dryad Digital Repository, 10.5061/dryad.hdr7sqvjr |

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
