## [Decision Letter]

**Acceptance summary:**

In this manuscript, Holt and colleagues investigate how the mechanoreceptor Piezo1 mediates keratinocyte cell migration and re-epithelialization during wound healing. The main findings are that loss of the Piezo1 protein in keratinocytes accelerates migration and wound healing, while genetic and pharmacological manipulations known to increase currents carried by Piezo1 slow migration and wound healing. The channels are shown to accumulate and cluster at the trailing edge of single migrating cells and at the wound margin during in vitro studies of wound healing. These findings demonstrate that Piezo1 mechanosensitive channels function as essential regulators of the speed of both migration and would healing. Furthermore, the findings suggest that increased flux through Piezo1 channels slows migration and wound healing. These channels are found to cluster in migrating cells and at wound margins. The conclusions are well-supported by the data and the authors' composition does an outstanding job of recognizing the limits of what has been learned and what remains uncertain.

**Decision letter after peer review:**

Thank you for submitting your article "Spatiotemporal dynamics of PIEZO1 localization controls keratinocyte migration during wound healing" for consideration by *eLife*. Your article has been reviewed by 3 peer reviewers, and the evaluation has been overseen by a Reviewing Editor and Kenton Swartz as the Senior Editor. The reviewers have opted to remain anonymous.

Essential revisions:

1. The controls used in Figure 1E and H appear to be different from each other which, makes it difficult to interpret the overall effect of the *Piezo1*-cKO or *Piezo1*-GoF genotypes. For example, the *Piezo1-*GoF keratinocytes in 1H do not appear to be very different from the WT control keratinocytes in 1E. This is also true of the *Piezo1-*cKO cells in 1E compared to the WT control cells in 1H. Why are these control groups so different? The reviewers are concerned that it may be difficult to draw conclusions about the deletion or enhancement of PIEZO1 in a set of animals, when the control variability is so great. Perhaps the background strain or something else about the animal contributes to the significant differences reported. Please comment on this.

2. There are some inconsistencies with the cohorts and controls used in certain experiments. In Figure 1J the authors utilize both the *Piezo1-*cKO and *Piezo1-*GoF cells for their in vivo experiments but when switching to an in vitro model they only test the *Piezo1-*cKO keratinocytes. It is not clear why the *Piezo1-*GoF cells are not included in Figure 1L to better correspond with the in vivo data. There is no rational given for why the authors instead utilized Yoda1. Additionally, it is unclear why there are so few keratinocytes in the Yoda1 experiment. In Figure 3D and E, it is unclear why the authors have not utilized the *Piezo1-*cKO keratinocytes as controls. The reviewers agree that the GoF mutants (ideally) or Yoda treatments with the appropriate controls should be added to many experiments, including Figure 1L, Figure 2, Figure 3 and Figure 4.

3. Related to the point above, the authors utilize 4 μM yoda1 throughout the manuscript however the rationale for this specific concentration is unclear. A Yoda1 concentration response experiment with WT and *Piezo1-*cKO keratinocytes would be beneficial in at least the first sets of Yoda1 experiments in order to determine if the observed effects are dependent on the level of PIEZO1 activation and to control for Yoda1 specificity.

4. The data shown in Figure 3 appear robust; however, it would be appropriate if authors quantify, compile, and compare the slope of the descending white lines denoting the velocity of retraction of independent experiments, for example. One suggestion is to plot the paired data (before and after addition of Yoda1) to demonstrate that indeed this increase in velocity happens most of the time in the wild-type but not in the cKO.

5. The experiment shown on Figure 4 appears to lack experimental control(s), especially because the results from it are used to derive the conclusion "PIEZO1-tdTomato enrichment and activity induces cellular retraction". Experiments could include inhibition of PIEZO1 and/or enhancement (chemically or genetically). Alternatively, the authors could monitor another excitatory cation channel, for instance, demonstrating that, unlike Piezo1, its function or localization does not correlate with cellular retraction.

6. The specific statistical tests and post hoc comparisons used in each data panel should be indicated for every figure in the figure legend. The authors should consider estimation statistics to evaluate the effect size along with conventional significance measures.

---

## [Author Response]

Essential revisions:1. The controls used in Figure 1E and H appear to be different from each other which, makes it difficult to interpret the overall effect of the *Piezo1-*cKO or *Piezo1-*GoF genotypes. For example, the *Piezo1-*GoF keratinocytes in 1H do not appear to be very different from the WT control keratinocytes in 1E. This is also true of the *Piezo1-*cKO cells in 1E compared to the WT control cells in 1H. Why are these control groups so different? The reviewers are concerned that it may be difficult to draw conclusions about the deletion or enhancement of PIEZO1 in a set of animals, when the control variability is so great. Perhaps the background strain or something else about the animal contributes to the significant differences reported. Please comment on this.

The Control for the cKO and GoF mutations are of different mouse backgrounds which likely is responsible for the different baseline activities. When performing any quantitative comparisons we always do so against the corresponding littermate Control mice. We have added the following sentences to the Results section (lines 120-124):

“A difference in the frequency of Ca^2+^ flickers between the Control_cKO_ and Control_GoF_ cells was observed, likely arising from different genetic backgrounds of the two strains. For this reason, in all subsequent experiments, mutant keratinocytes are compared to littermate Control cells of the same genetic background.”

Additionally, we have clarified the background of the Control samples throughout the manuscript using the notation Con_cKO_ and Con_GoF_.

The information about the mouse backgrounds is now also added to the Methods section, Animals subsection (lines 643-645):

“Piezo1^fl/fl^ mice were generated in C57BL/6 background and Piezo1^cx/cx^ mice were initially generated in BALB/c background and then maintained in C57BL/6 for > 10 generations. K14^Cre^ mice were in the C57BL/6.”

2. There are some inconsistencies with the cohorts and controls used in certain experiments. In Figure 1J the authors utilize both the *Piezo1-*cKO and *Piezo1-*GoF cells for their in vivo experiments but when switching to an in vitro model they only test the *Piezo1-*cKO keratinocytes. It is not clear why the *Piezo1-*GoF cells are not included in Figure 1L to better correspond with the in vivo data. There is no rational given for why the authors instead utilized Yoda1. Additionally, it is unclear why there are so few keratinocytes in the Yoda1 experiment. In Figure 3D and E, it is unclear why the authors have not utilized the *Piezo1-*cKO keratinocytes as controls. The reviewers agree that the GoF mutants (ideally) or Yoda treatments with the appropriate controls should be added to many experiments, including Figure 1L, Figure 2, Figure 3 and Figure 4.

Data from some control conditions could not be included in the original submission due to pandemic-related unavailability of mouse samples. In the revised manuscript we have included additional data as requested:

Figure 1:

– We have included data from Piezo1-GoF keratinocytes and corresponding littermate Controls for in vitro wound healing experiments. Consistent with our in vivo results, we observe that monolayers of *Piezo1*-GoF keratinocytes close scratches significantly slower compared to those made in littermate control cells. These results have been included in revised Figure 1L (*middle*) and the Results section (lines 136-137) has been updated to include “Conversely, scratch closure in monolayers of *Piezo1*-GoF keratinocytes was significantly slower (Figure 1L, *middle*).”

– We have increased the number of keratinocyte scratch assays treated with Yoda1 and these results can be found in updated Figure 1L (*right*).

– We have also included experiments utilizing *Piezo1-*cKO keratinocytes as controls for Yoda1 treatment (Figure 1—figure supplement 3B) and included this sentence in the revised manuscript (lines 140-142): “No effect on wound closure was observed when *Piezo1*-cKO monolayers were treated with Yoda1 indicating that inhibition of scratch closure is the result of PIEZO1 (Figure 1—figure supplement 3B).”

Figure 2:

– We have included data from the *Piezo1*-GoF keratinocytes for the single cell tracking assay. We observed that *Piezo1*-GoF cells migrated slower than littermate controls. They also migrate straighter than littermate controls, resulting in no marked difference in the MSD. These results can be found in updated Figure 2 and Figure 2—figure supplements 2 and 3, and Figure 2—Video 2. We revised the Results section to include these new results (lines 185-190):

“Similarly, we performed the same experiments and extracted cell migration trajectories from single keratinocytes harvested from *Piezo1*-GoF and littermate Control_GoF_ mice (Figure 2—figure supplement 2, Figure 2—Video 2). We observed no difference in the MSD plots of *Piezo1-*GoF keratinocytes and littermate control cells (Figure 2—figure supplement 3). However, separating the data into directionality and speed, *Piezo1-*GoF cells moved straighter (Figure 2—figure supplement 3) and slower (Figure 2E).”

Figure 3 (Figure 5 in the revised manuscript):

– We have included results from *Piezo1-*GoF keratinocytes to visualize edge dynamics and changes in cell position over time. This has been included for both single cell migration (Figure 5C-D, Figure 5—figure supplement 2 and Figure 5—Video 4) as well as for wound-healing monolayers (Figure 5—figure supplement 3 and Figure 5—Video 7). We have revised the manuscript text with:

For data from single GoF keratinocytes (lines 343-345):

“Additionally, kymographs of *Piezo1-*GoF keratinocytes also showed an increase in cell edge dynamics compared to littermate Control_GoF_ cells further supporting PIEZO1’s role in retraction (Figure 5—figure supplement 2 and Figure 5—Video 4).”

For data from GoF keratinocyte monolayers (lines 371-374):

“In *Piezo1-*cKO monolayers we observed cells protrude forward into the cell-free space to close scratch wounds to a greater extent than controls (Figure 5H, Figure 5—Video 6), while GoF monolayers did so to a lower extent (Figure 5—figure supplement 3, Figure 5—Video 7).”

– We have also included the *Piezo1*-cKO wound-healing monolayers as controls in our updated manuscript, without (Figure 5H) and with Yoda1 (Figure 5—figure supplement 4, Figure 5-Video 3). We observed that the cKO monolayers migrate into the wound area faster than controls and Yoda1 has no effect on this. We have updated the Results section (lines 371-376) by stating:

“In *Piezo1-*cKO monolayers we observed cells protrude forward into the cell-free space to close scratch wounds to a greater extent than controls (Figure 5H, Figure 5—Video 6), while GoF monolayers did so to a lower extent (Figure 5—figure supplement 3, Figure 5—Video 7). Additionally, no effect of Yoda1 addition was seen on the rate of advancement in kymographs taken at the wound edge of *Piezo1-*cKO monolayers (Figure 5—figure supplement 4).”

Figure 4 (split between Figures 3 and 4 in the revised manuscript)

– Our results demonstrating PIEZO1-tdTomato enrichment at the rear of single migrating cells suggest that PIEZO1 may underlie cell polarization during migration. In the revised manuscript we have investigated this possibly by using unbiased cellular morphometrics. Based on the observed localization pattern of PIEZO1-tdTomato, we hypothesized that *Piezo1-*cKO and *Piezo1-*GoF keratinocytes would display different cell shapes compared to littermate controls. We find that knocking out PIEZO1 reduced cell polarization while the GoF PIEZO1 increased the occurrence of highly polarized shapes. These results are now included in Figure 3, and the revised manuscript includes this description (lines 201-217):

“This observation of PIEZO1-tdTomato enrichment at the rear of single migrating cells suggest that PIEZO1 may underlie cell polarization during migration.

To determine whether PIEZO1 may be responsible for generating the polarized shape, we performed cellular morphometrics on the single cell images obtained from the above time-lapse imaging of single cell migration. We used visually-aided morpho-phenotyping recognition (VAMPIRE) (Phillip et al., 2021), a high-throughput machine-learning algorithm that analyzes the morphology of individual cells in a population by quantifying shape modes of segmented cells and showing the level of correlation between the shape modes through a dendrogram (Figure 3 B, C). VAMPIRE classification of the *Piezo1*-cKO and littermate Control_cKO_ keratinocytes into 20 shape modes revealed that *Piezo1*-cKO reduced the proportion of highly polarized shapes and increased the proportion of weakly polarized shapes relative to littermate Control_cKO_ keratinocytes (Figure 3D, E). On the other hand, the GoF mutation increased the frequency of polarized and highly polarized shapes at the expense of unpolarized or weakly polarized cell shapes (Figure 3F, G). Taken together, these results indicate that PIEZO1 activity promotes cell polarization. Based on imaging the localization of endogenous PIEZO1 channels in migrating cells, it appears that this may be mediated by regulation of the channel’s subcellular localization.”

3. Related to the point above, the authors utilize 4 μM yoda1 throughout the manuscript however the rationale for this specific concentration is unclear. A Yoda1 concentration response experiment with WT and *Piezo1-*cKO keratinocytes would be beneficial in at least the first sets of Yoda1 experiments in order to determine if the observed effects are dependent on the level of PIEZO1 activation and to control for Yoda1 specificity.

As suggested by the reviewers, we performed a Yoda1 concentration curve using Control_cKO_ and *Piezo1-*cKO keratinocytes and have included the results in new Figure 1—figure supplement 3. We observe that Yoda1 has an effect on inhibiting wound closure at concentrations greater than 2 μM and we have stated this in the updated Results section. We also confirmed the specificity of Yoda1 by treating *Piezo1-*cKO monolayers with the highest concentration we tested, 4μM.

The Results section of the manuscript (lines 137-142) has been updated with these results by adding that:

“Correspondingly, when the PIEZO1 agonist Yoda1 was added to healing Control_cKO_ monolayers at concentrations greater than 2 μM, scratch wound closure was also significantly impaired (Figure 1L, right, Figure 1—figure supplement 3A), further supporting PIEZO1 involvement in re-epithelialization. No effect on wound closure was observed when *Piezo1-*cKO monolayers were treated with Yoda1 indicating that inhibition of scratch closure is the result of PIEZO1 (Figure 1—figure supplement 3B).”

4. The data shown in Figure 3 appear robust; however, it would be appropriate if authors quantify, compile, and compare the slope of the descending white lines denoting the velocity of retraction of independent experiments, for example. One suggestion is to plot the paired data (before and after addition of Yoda1) to demonstrate that indeed this increase in velocity happens most of the time in the wild-type but not in the cKO.

(Figure 3 from the previous version is now Figure 5 in the revised manuscript)

As the reviewers suggested, we quantified and compared the slope of the descending white lines denoting the velocity of retraction in the shown kymographs. This quantification can be found in Figure 5—figure supplement 1C. Since this quantification only represents behavior of one point along a single cell for each condition, we performed additional analyses to more objectively and rigorously quantify PIEZO1’s effect on cellular retraction. For this, we segmented migrating keratinocytes from the DIC timelapse videos. Using these segmented images we then used the open source software ADAPT to quantify cellular morphodynamics before and after Yoda1 addition in Control_cKO_, and in *Piezo1-*cKO, Control_GoF_ and *Piezo1-*GoF cells. This method allows for a robust and unbiased approach as it quantifies the velocity at every point along the detected cell boundary at every time point of a time lapse series. This quantification supports and greatly strengthens our initial observation linking PIEZO1 activity to cell edge velocity and retraction rates.

We have included these results in new Figure 5C-E and the Results section (lines 346-362) has been updated with the addition of:

“Kymograph quantitation is limited to one point along the cell edge. To more objectively investigate the effect that PIEZO1 activation has on cell morphodynamics, cells were segmented for each frame of DIC time-lapse series for the following conditions: Control_cKO_ cells before and after Yoda1 addition, *Piezo1-*cKO cells, Control_GoF_ cells and *Piezo1-*GoF keratinocytes (Figure 5C). By comparing segmented outlines between frames we could obtain the velocity of the cell edge at every position along the detected boundaries for protrusion events (positive velocities) and retraction events (negative velocities). Yoda1 treatment resulted in a significant increase in edge velocities relative to DMSO-treated Control_cKO_ cells (Figure 5D). In contrast, *Piezo1-*cKO keratinocytes showed a reduction in edge velocities relative to litter-mate Control_cKO_ cells. Yoda1, which is expected to globally activate PIEZO1 channels, resulted in an increase of cell edge velocity during both protrusion and retraction events, though the increase in retraction velocity was greater. Heatmaps of cell edge velocity illustrate the robustness of this response (Figure 5E). Consistent with our observations of Yoda1 treatment, *Piezo1-*GoF keratinocytes also showed a significant increase in edge velocities relative to littermate Control_GoF_ (Figure 5D, Figure 5—figure supplement 2). In addition, there was a clear increase in the proportion of retracting positions relative to protruding positions in GoF keratinocytes. These results reveal that PIEZO1 activity regulates cell edge dynamics and further support our observations that PIEZO1 activity increases cellular retraction.”

5. The experiment shown on Figure 4 appears to lack experimental control(s), especially because the results from it are used to derive the conclusion "PIEZO1-tdTomato enrichment and activity induces cellular retraction". Experiments could include inhibition of PIEZO1 and/or enhancement (chemically or genetically). Alternatively, the authors could monitor another excitatory cation channel, for instance, demonstrating that, unlike Piezo1, its function or localization does not correlate with cellular retraction.

*– The experiment shown on Figure 4 appears to lack experimental control(s):* We realize that our description of this section may have been unclear. For these measurements, we ask: do regions displaying PIEZO1-tdTomato enrichment migrate differently from regions without enrichment? Thus, the regions without PIEZO1-tdTomato enrichment serve as Controls for regions displaying PIEZO1-tdTomato enrichment. By comparing monolayer edge advancement in the Control and enriched regions, we can directly examine the relationship between channel enrichment and monolayer edge retraction. We have now revised the text in the Results section (lines 274-280) to be clearer:

“We asked whether regions displaying PIEZO1-tdTomato enrichment migrate differently from regions without enrichment. To systematically assess the relationship between PIEZO1-tdTomato enrichment and wound edge dynamics, we used kymographs to graphically represent PIEZO1-tdTomato position over the imaging period from regions that displayed PIEZO1-tdTomato enrichment at the wound edge (Figure 4 C, 4E,Figure 4—Video 1) and compared them to control regions that showed no such channel enrichment throughout the videos (Figure 4B, 4D, Figure 4—Video 2).”

*– Authors could monitor another excitatory cation channel, for instance, demonstrating that, unlike Piezo1, its function or localization does not correlate with cellular retraction:* The experiments visualizing PIEZO1 localization were made possible by the existence of the PIEZO1-tdTomato mouse (Ranade et al., PNAS 2014), wherein the endogenous PIEZO1 channel is fused to a tdTomato reporter protein. Performing an analogous experiment for other cation channels would require first investigating the expression of candidate channels in keratinocytes, and then obtaining or making an engineered mouse, harvesting keratinocytes, and imaging channel localization over several hours. These are challenging studies that we believe are beyond the scope of this study. Additionally, the fact that PIEZO1 localization correlates with cellular retraction does not necessarily preclude the contributions of other channels.

*– Experiments could include inhibition of Piezo1 and/or enhancement (chemically or genetically):* We would have really liked to image PIEZO1-tdTomato dynamics in healing monolayers in the presence of Yoda1 or GsMTx-4 as suggested by the reviewers. However, determining whether enrichment events are increased or decreased in drug-treated healing monolayers would be extremely difficult to interpret due to the stochastic nature of enrichment dynamics and the long imaging times. Appropriate interpretation of the data would require quantitation of channel localization dynamics through a custom-built image analysis pipeline, which would have been a major feat outside of the scope of this study.

We agree with the reviewers that our results from Figure 4 alone do not support the conclusion "PIEZO1-tdTomato enrichment and activity induces cellular retraction". To more precisely summarize the insights from revised Figures 4 and 5 we have replaced the sentence with (lines 379-382):

“Taken together, our results demonstrate that PIEZO1 induces cellular retraction to slow single and collective cell migration and thus causes delayed wound healing. We propose that dynamic enrichment of the channel protein serves to locally amplify channel activity and the downstream retraction events.”

6. The specific statistical tests and post hoc comparisons used in each data panel should be indicated for every figure in the figure legend. The authors should consider estimation statistics to evaluate the effect size along with conventional significance measures.

We have modified each figure legend to state which statistical test was used in every data panel. Following the recommendation of the reviewers, we have transitioned to using estimation statistics and Cumming plots to represent our data and visualize the effect size of our results. We generated these plots using the freely available software on https://www.estimationstats.com/#/ based on the publication from Ho et al., 2019 (DOI: 10.1038/s41592-019-0470-3). These plots show both groups plotted on the upper raw data axes. The raw data is aligned vertically with a forest plot allowing for the visual and statistical comparison of the Cohen’s d for each condition. We have also stated the exact Cohen’s *d* value and *p* values in each legend description.

We have revised the Statistical Analysis subsection of the Methods section (lines 829-839) to describe these changes:

“Statistical Analysis

Sample sizes are indicated in corresponding figures. Cumming estimation plots were generated and Cohen’s *d* in all plots (except Figure 5D) was calculated using an online estimation stats tool (https://www.estimationstats.com). Estimation plots show the raw data plotted on the upper axes with bars beside each group denoting the sample mean ± s.d.; the mean difference and Cohen’s *d* effect size is plotted on the lower axes. On the lower plot, the mean difference is depicted as a dot; the 95% confidence interval is indicated by the ends of the bold vertical error bar. OriginPro 2020 (OriginLab Corporation) was used for calculating *p* values (for all figures except for Figure 5D) and generating plots used in Figure 1B, 2B-D, 3D-G, 4G, 5D, Figure 1—figure supplement 2 and Figure 2—figure supplements 1-3. All plots generated in OriginLab are presented as the mean ± SEM. Statistical tests used to calculate *p* values are indicated in figure legends. For Figure 5D, a Python script was used for calculating Cohen’s *d* and *p* values.”

In view of the extensive new results included in our resubmitted manuscript we have also revised the Discussion section.